# Mitochondrial VDAC1: A Potential Therapeutic Target of Inflammation-Related Diseases and Clinical Opportunities

**DOI:** 10.3390/cells11193174

**Published:** 2022-10-10

**Authors:** Hang Hu, Linlin Guo, Jay Overholser, Xing Wang

**Affiliations:** 1Inflammation & Allergic Diseases Research Unit, The Affiliated Hospital of Southwest Medical University, Luzhou 646000, China; 2Department of Respiratory and Critical Care Medicine, The Affiliated Hospital of Southwest Medical University, Luzhou 646000, China; 3Department of Obstetrics and Gynecology, The Ohio State University Wexner Medical Center at The Ohio State University, Columbus, OH 43210, USA

**Keywords:** VDAC1, inflammation, mitochondria, metabolism, Ca^2+^, mitophagy

## Abstract

The multifunctional protein, voltage-dependent anion channel 1 (VDAC1), is located on the mitochondrial outer membrane. It is a pivotal protein that maintains mitochondrial function to power cellular bioactivities via energy generation. VDAC1 is involved in regulating energy production, mitochondrial oxidase stress, Ca^2+^ transportation, substance metabolism, apoptosis, mitochondrial autophagy (mitophagy), and many other functions. VDAC1 malfunction is associated with mitochondrial disorders that affect inflammatory responses, resulting in an up-regulation of the body’s defensive response to stress stimulation. Overresponses to inflammation may cause chronic diseases. Mitochondrial DNA (mtDNA) acts as a danger signal that can further trigger native immune system activities after its secretion. VDAC1 mediates the release of mtDNA into the cytoplasm to enhance cytokine levels by activating immune responses. VDAC1 regulates mitochondrial Ca^2+^ transportation, lipid metabolism and mitophagy, which are involved in inflammation-related disease pathogenesis. Many scientists have suggested approaches to deal with inflammation overresponse issues via specific targeting therapies. Due to the broad functionality of VDAC1, it may become a useful target for therapy in inflammation-related diseases. The mechanisms of VDAC1 and its role in inflammation require further exploration. We comprehensively and systematically summarized the role of VDAC1 in the inflammatory response, and hope that our research will lead to novel therapeutic strategies that target VDAC1 in order to treat inflammation-related disorders.

## 1. Introduction

Inflammation is a defense response of the body to stimuli, such as infectious and non-infectious triggers. Inflammation can be beneficial when it occurs in moderation; however, excessive inflammation can easily become detrimental events that result in possible damage to local tissues. In understanding the mechanism of chronic inflammation, we know that it has a deep relationship with various diseases, for example, type 2 diabetes, atherosclerosis, asthma, neurodegenerative diseases, cancers and others [1,2,3].

Mitochondria are vital organelles in eukaryotic cells. They are not only involved in oxidative phosphorylation, thermogenesis, the biogenesis of iron–sulfur clusters, and in heme, lipid and amino acid biosynthesis [4,5,6], they can modulate programmed cell death [7,8] and control inflammation [9]. Mitochondrial malfunction is related to various diseases [10,11,12] that are mainly manifested with a reduction in metabolism, Ca^2+^ homeostatic imbalance, increased levels of reactive oxygen species (ROS), lipid peroxidation and increased apoptosis (Figure 1).

Voltage-dependent anion-selective channel protein was first purified from paramecium mitochondria in 1976 [13]. We now know that there are two isoforms of voltage-dependent anion channel (VDAC) in yeast, yVDAC1 and yVDAC2, with yVDAC1 being the most abundant [14,15]. Three VDAC family members in mammalian mitochondria were observed, VDAC1, VDAC2, VDAC3. VDAC1 is the most widely expressed, and contributes to a broad and general role [16,17,18]. Notably, VDAC2 in mammals contributes to anti-apoptotic phenotypes by binding to Bcl-2 homologous antagonist killer (BAK); mitochondrial apoptosis is activated, resulting from the homo-oligomerization of BAK when VDAC2 is displaced by truncated BH3 interacting-domain death agonist (tBID), Bcl-2-like protein 11 (BIM) or Bcl-2-associated agonist of cell death (BAD) [19]. VDAC3, especially the indispensable cysteine residues, plays an important role in protecting mitochondria from oxidative stress [20]. The transcriptional factors that regulate cell growth, apoptosis, energy metabolism, etc., also regulate VDAC gene expressions [21]. More information on VDAC isoforms and gene regulation has been documented by Vito De Pinto et al. [18,21]. In this review, we focus only on the most abundant isoform, VDAC1, and its relationship with inflammation. 

The 3-dimensional structure of VDAC1 shows that the 19 transmembrane β-strands form a flexible loop, forming a β-pore containing a 25-residue segment in the N-terminal domain. The migration of the N-terminal domain is involved in channel gating, and in the formation of VDAC1 dimers that transport metabolites and molecules to maintain mitochondrial homeostasis [16,22,23]. This VDAC1 structure was published by three independent groups in 2008 [24,25,26]. The β1 (^26^Leu-Ile-Lys-Leu-Asp-Leu-Lys-Thr-Lys-Ser^35^) and β19 (^273^His-Lys-Leu-Gly-Leu-Gly-Leu-Glu-Phe-Gln^282^) strands are parallel [24,25]. Bcl-2 protein Bcl-xL interacts with β17 (^243^Ile-Gly-Leu-Gly-Tyr-Thr-Gln-Thr-Leu^251^) and β18 (^255^Ile-Lys-Leu-Thr-Leu-Ser-Ala-Leu-Leu^263^), fulfilling an anti-apoptotic function by suppressing mitochondrial release apoptogenic proteins [25]. The conserved and flexible sequence (^21^Gly-Tyr-Gly-Phe-Gly^25^) acts as a bridge that connects the α-helix to β1 [26]. The α-helix is at the midway point of the barrel pore in a horizon position [24]; it acts as a gate by narrowing the pore cavity to modulate metabolite transportation [26]. 

VDAC1 is a multifunctional channel protein that is located in the outer membrane of mitochondria. It modulates cellular metabolism [25,27]. VDAC1 regulates metabolism between the mitochondria and other parts of the cell by transferring metabolites, such as pyruvate, malonate, succinate, nucleotides and nicotinamide adenine dinucleotide hydrogen (NADH), into the mitochondria to complete subsequent metabolic reactions [27]. VDAC1 is also involved in cholesterol transportation, regulating lipid metabolism, mediating ion channels, regulating Ca^2+^ signaling between mitochondria and the endoplasmic reticulum (ER), and regulating the redox status of mitochondria and the cytoplasm. It has also been suggested that VDAC1 is a key protein that is involved in mitochondria-induced cell death [28,29,30,31].

VDAC1 is associated with increased release of mitochondrial DNA (mtDNA) [32,33], which is a signal of impaired mitophagy [34]. Mitophagy plays a central role in maintaining mitochondrial homeostasis; the process is pivotal in the development of inflammation and apoptosis [35,36,37,38], and is highly related to cytokines release [38,39]. VDAC1 plays an important role in regulating the mitochondrial involvement in vital activities (Figure 1). Functional abnormalities in mitochondria may lead to mitochondria-derived pathologic diseases, including inflammation, cardiovascular disease, cancer, neurodegenerative diseases, diabetes, and so on [10,11,40,41]. 

VDAC1 may become a potential therapeutic target and a breakthrough for many diseases. Our current knowledge of VDAC1 is insufficient. It is urgent to carry out further explorations on the molecular mechanisms of VDAC1, which may hopefully lead to novel treatment strategies for inflammation-related diseases.

## 2. Inflammation, VDAC1 Mediates Apoptosis and Mitochondrial Oxidative Stress

Programmed cell death is associated with many different kinds of inflammatory diseases, and is a major determinant of inflammatory disease severity [42]. Many studies have revealed key pathological mechanisms of apoptosis that are involved in infectious and inflammatory diseases. Sepsis-derived lymphopenia and immunosuppression are associated with the apoptosis of lymphocytes and parenchymal tissues [43]. It has been also indicated that apoptotic inflammatory cells may play an important role in the development of inflammation [44,45,46,47,48]. Apoptosis of inflammatory microvascular cells may lead to dysregulation of microvascular repair and damage that result from a malfunction in endothelial cells, and cause diseases [49].

### 2.1. VDAC1 Regulates Inflammation via Mediating Apoptosis

The mitochondrial permeability transition pore (MPTP) is about 1.4 nm in diameter, and supports solute and ion diffusion under 1500 kDa. It is also known as the mitochondrial macro-channel that plays an important role in cell survival and apoptosis [50,51]. The voltage-dependent anion channel (VDAC) is located in the outer mitochondrial membrane (OMM); adenine nucleotide translocase (ANT) is located in the inner mitochondrial membrane (IMM). VDAC and ANT are considered to be the structural components of the MPTP [52,53,54]. 

The Bcl-2 family has a close relationship with mitochondria and apoptosis [55]. It is known that Bcl-2 family member, Bcl-2-associated X protein (BAX), interacts with VDAC1 to regulate the release of cytochrome c (Cyto c) during apoptosis [56,57] (Figure 2). Oligomerization of BAX is one of the mechanisms that is involved in the mitochondrial apoptosis pathway [58]. A rat brain model indicates BAX promotes apoptosis by interacting with VDAC1 to expand the associated pore size, resulting in the increased permeability of mitochondria [59]. During apoptosis, VDAC1 assembles into oligomeric structures, forming a channel that is sufficient to pass Cyto c and release it into the cytoplasm. Cyto c forms oligomeric apoptosomes by binding to Apaf-1, apostasy activator and deoxyadenosine triphosphate (dATP); this results in the activation of cysteine protease 9 (caspase-9) that further activates effector caspases, caspase-3, caspase-6 and caspase-7 [55,60,61]. Ultimately, the caspase-mediated apoptosis pathway proteolytic cascade begins to cleave organelles and cellular components, resulting in apoptosis [60].

Apoptosis may cause mitochondrial outer membrane permeabilization (MOMP), which further induces inflammatory responses via multiple pathways [62] (Figure 2). The outer mitochondrial membrane pore gradually enlarges after MOMP, further causing extrusion and rupture of the inner mitochondrial membrane (IMM); this leads to the release of mitochondrial DNA (mtDNA) into the cytoplasm [62] (Figure 2). The cytoplasmic mtDNA, together with cyclic GMP-AMP synthase (cGAS)-stimulator of interferon genes (STING), signal the release of pro-inflammatory interferon signal [63,64]. Additionally, MOMP can induce proteasomal degradation of inhibitors of apoptosis (IAPs), which leads to nuclear factor-κB (NF-κB)-induced kinase (NIK) to further induce pro-inflammatory NF-κB signaling as well as activated caspase-8, which in turn results in the maturation of pro-inflammatory factor interleukin 1β (IL-1β) [62].

### 2.2. VDAC1 Mediates Mitochondrial Oxidative Stress in Immune Responses

ROS from mitochondria can be dramatically induced under the stimulation of radiation, cigarette smoke, air pollution, inflammatory factors, tumor necrosis factor, hyperlipidemia, hypoxia, and so on. Notably, the ROS are mainly generated from the respiratory complex that is located in the IMM [65,66]. Malfunctioning mitochondria hyperproduce ROS which negatively affect other components of mitochondria, for example, mtDNA, membrane lipids, oxidative phosphorylation, etc. [67,68]. The mtDNA is mainly localized in the IMM, and mtDNA can easily be oxidized by ROS to generate oxidized mtDNA fragments (fomtDNA) [69]. The released mtDNA acts as ligands for different danger signal sensors, activating the innate immune response (Figure 2). These risk sensors include the cytoplasmic cyclic GMP-AMP synthase (cGAS); Toll-like receptor 9 (TLR9); nucleotide-binding domain and leucine-rich repeat (LRR) containing P3 (NLRP3) inflammasome; and absent in melanoma 2 (AIM2) inflammasome [65]. Through these pathways, mtDNA can induce the secretion of inflammatory cytokines, and induce the recruitment of immune cells at different sites, providing the conditions for inflammation in many diseases [65]. It has been shown that VDAC1 oligomer pores promote MOMP and allow the release of mtDNA into the cytoplasmic matrix in living cells, where mtDNA fragments escape from the mitochondria through direct interactions at the N-terminus of VDAC1 [32]. At the same time, the inhibition of VDAC1 oligomerization eliminates cytoplasmic and circulating mtDNA. Therefore, single-stranded or double-stranded DNA escapes into the cytoplasm through the permeability transition pore that is composed of VDAC1. VDAC1 indirectly participates in mtDNA induction by mediating the translocation of the subsequent mtDNA inflammatory response [32]. VBIT-3 and VBIT-4, as well as VBIT-12, were reported to interact with VDAC1 by disrupting its oligomerization, resulting in altered intracellular Ca^2+^ concentration and decreased ROS levels, thereby protecting mitochondrial malfunction related to apoptosis and inflammation [70,71]. This response was found to alleviate type 2 diabetes [72], lupus [32], atrial myocardium fibrosis [73], ulcerative colitis [74] and amyotrophic lateral sclerosis [71]. Additionally, silencing VDAC1 in cancer cells can suppress tumor cell proliferation, in vivo and in vitro [75,76]. Silencing VDAC1-enhanced mitochondrial function and synaptic activity provides a potential therapeutic approach for neuron-related diseases (Alzheimer’s, etc.) [77,78]. Increasing evidences indicates that targeting VDAC1 with small molecules may be worth further investigation since it may provide novel strategies against diseases that are associated with mitochondrial disorder.

The cGAS-STING pathway mediates the escape of mtDNA from stressed mitochondria, provoking inflammation and further leading to calcium uptake and the triggering of VDAC oligomerization [33]. The cGAS is a newly discovered sensor that serves as a hazard-associated molecular pattern for the detection of cytoplasmic mtDNA [79]. The mtDNA binds to cGAS in a sequence-independent manner, and then induces a conformational change in the catalytic center of cGAS; this allows the enzyme to convert GTP and ATP into the second messenger, cyclic GMP-AMP. Cyclic GMP-AMP is a molecule that high-affinity gametes of STING subsequently recruit and activate TANK-binding kinase 1 (TBK1) and interferon regulatory factor 3 (IRF3) through a phosphorylation-dependent mechanism [80]. STING also activates NF-κB, which together with IRF3, turns on the transcription of type I interferon (IFN) and other cytokines [81], forming the basis for subsequent inflammatory responses. 

Toll-like receptor 9 (TLR9) is a cellular DNA receptor of the innate immune system. It plays a key role in the immune inflammatory response [82,83]. TLR9 is expressed as a homodimeric complex on the inner surface of the endosomal membrane. TLR9 is activated by unmethylated CpG sequences that are present in DNA molecules, including mtDNA; it binds specifically to the N-terminus of the C-shaped leucine-rich repeat region of TLR9 through mitogen-activated protein kinase (MAPK). The NF-κB pathway of activated B cells interacts with the MyD88 adaptor protein, leading to the transcription of inflammatory cytokines, such as tumor necrosis factor-α (TNF-α), IL-6 and IL-12, thereby activating the inflammatory response [83,84].

As an important component of innate immunity, the NLRP3 inflammasome plays an important role in the body’s immune response to disease occurrence [85]. NLRP3 can be bound by oxidized mtDNA that is released during apoptosis [86]. Although the exact mechanism is unclear, evidence has suggested that mtDNA is essential for NLRP3 signaling. For example, autophagy that eliminates damaged mitochondria prevents inflammasome activation [87]; drugs that inhibit mtDNA synthesis also inhibit NLRP3 inflammasome activation [86]. The reintroduction of oxidized DNA into macrophages restores the inhibition of mtDNA synthesis NLRP3 activation [88]. The formation of oligomerized VDAC1 is associated with mtDNA [32,33]. cGAS-STING signaling mediates the oxidization of mtDNA that binds with cytosolic NLRP3, in which inflammasome activators stimulate calcium uptake to open mitochondrial permeability transition pores (mPTP) and trigger VDAC1 oligomerization [33].

AIM2 is a type of innate immune sensor that detects altered or misplaced DNA molecules, such as damaged DNA and DNA that is abnormally present in the cytoplasmic compartment [89,90,91,92]. After binding to DNA, AIM2 assembles a multiprotein innate immune complex called the inflammasome, which can lead to the activation of inflammatory caspases, resulting in the maturation and secretion of cytokines IL-1β and IL-18. AIM2 can also trigger pyroptosis, a pro-inflammatory form of cell death [90]. Recent studies have shown that the detection of self-DNA by AIM2 is an important factor in diseases that are associated with disturbances in cellular homeostasis [90,92]. Taken together, targeting VDAC1 channels in order to reduce apoptosis and mitochondrial oxidative stress may provide new solutions for treating inflammatory diseases.

## 3. Inflammation and VDAC1 Mediates Mitochondrial Ca^2+^ Transportation

Mitochondrial Ca^2+^ uptake and release play a key role in cellular physiology by regulating intracellular Ca^2+^ signaling, energy metabolism and cell death [93]. The transportation of Ca^2+^ across the inner or outer mitochondrial membranes (IMM, OMM) is mainly mediated by several proteins, including VDAC1, mitochondrial Ca^2+^ monotransporter (MCU) and Na^+^-dependent mitochondrial Ca^2+^ efflux transporter (NCLX) [94,95]. 

VDAC1 was shown to be highly permeable to Ca^2+^, and contains a binding site for ruthenium red, thereby inhibiting channel opening [96,97]. VDAC1 may be a key component of the mitochondrial Ca^2+^ homeostatic mechanism, enhancing the Ca^2+^ response through different mechanisms. VDAC1 acts as a large conductance channel that allows for the rapid diffusion of Ca^2+^ across the OMM, thereby allowing the exposure of low-affinity single transporters in the inner membrane to the high Ca^2+^ microdomains that are generated by the opening of the endoplasmic reticulum (ER)-Ca^2+^ channel [96,97]. 

Moreover, VDAC1 has been proposed to be part of a larger complex of members, including the adenine nucleotide transporter, cyclophilin D, peripheral benzodiazepine receptors and Bcl-2 family members [98], which can interact with the ER. The structural components interact with each other, and thus become part of the molecular mechanism of mitochondrial docking with Ca^2+^. VDAC1 is the major permeation pathway for Ca^2+^ across the OMM, and VDAC1 mediates Ca^2+^ transport through the OMM to the IMM space. It can also facilitate Ca^2+^ transport from the inner mitochondrial membrane space (IMS) into the cytoplasm [94] (Figure 3A). 

Ca^2+^ is an important regulatory point of barrier function and inflammation. Ca^2+^ influx is involved in many steps of the inflammatory cascade, including leukocyte rolling, arrest, adhesion, and ultimately, transendothelial migration, etc. [99]. Ca^2+^ is involved in lymphocytic responses to foreign antigens, and inositol triphosphates (InsP3) are generated as a result of foreign molecules binding to antigen receptors and stimulating Ca^2+^ release from internal storage [100]. Once these stores are emptied, store-operated Ca^2+^ channels (SOCs) are activated, allowing lymphocytes to maintain a long-term increase in Ca^2+^; this usually occurs in the form of a series of regular Ca^2+^ oscillations that activate nuclear factor of activated T cells (NF-AT) [100]. Studies have shown that increased mitochondrial calcium levels promote the activation of CD4 T cells [101,102]. Immunosuppressants, such as cyclosporine, function by inhibiting Ca^2+^-dependent activation of NF-AT; this also emphasizes the importance of Ca^2+^ signaling in immune cell activation.

### 3.1. Neutrophils

Neutrophils are the most abundant type of white blood cells, and are the first responders to inflammatory stimuli, such as bacterial infection, or tissue damage medium caused by polarization and migration of mediators such as formyl-Met-Leu-Phe (fMLP) and IL-8 [103,104], whose dysfunction often leads to severe infections and inflammatory autoimmune diseases. 

In neutrophils, the cytoplasmic free calcium concentration is an important determinant of cell viability, and is a marker of neutrophil activation; it is closely related to a range of neutrophil functions [105]. Rapid cell spreading in neutrophils is induced by Ca^2+^ signaling [106]; Ca^2+^ influx activates cytoplasmic calpain, which plays an important role in regulating neutrophil polarization, and in directing their migration toward chemotactic stimuli [103]. The entry of extracellular Ca^2+^ into neutrophils affects multiple functions, including phagocytosis, ROS production, vesicle secretion and degranulation, β2-integrin activation, and cytoskeletal rearrangement that leads to polarization and migration; these activities play a key role in the occurrence and development of the neutrophil inflammatory response [107,108] (Figure 3B).

### 3.2. Macrophages

Ultrasound, combined with endogenous protoporphyrin IX derived from 5-aminolevulinic acid (ALA-SDT), induce the apoptosis of macrophages [109]. The inhibition of VDAC1 by 4,4′-diisothiocyanostilbene-2,2′-disulfonic acid (DIDS) was found to prevent ALA-SDT-induced cell apoptosis in THP-1 macrophages [109]. The VDAC1 of the mycobacterium avium phagosome is associated with bacterial survival and lipid export in macrophages [110]. Macrophages are an important part of the innate immune system; their main function is to phagocytose and digest cell debris and pathogens, and they play an important role in inflammatory responses [111,112]. 

Ca^2+^ is an essential second messenger in phagocytosis; indeed, elevated cytosolic calcium concentrations are required for efficient phagocytosis and maturation of phagosomes [113]; blocking MCU inhibits macrophage phagocytosis [114]. Studies have shown that Ca^2+^ influx into macrophages is the trigger for macrophage activation, and that increased Ca^2+^ concentrations are associated with macrophage differentiation [115]. An influx of extracellular Ca^2+^ is required to polarize macrophages toward the pro-inflammatory M1 phenotype, while decreasing Ca^2+^ leads to anti-inflammatory M2 switching [114,116] (Figure 3B).

### 3.3. Dendritic Cells

Dendritic cells (DCs) are the most powerful antigen-presenting cells in the body. DCs uptake, process and present antigens efficiently that are crucial for initiating T cell responses. They play a central role in initiating, regulating and maintaining immune responses [117]. 

Ca^2+^ signaling plays a key role in the function of DCs. Migration of DCs to secondary lymphoid organs is indispensable for subsequent T helper cell-mediated adaptive immunity. It has been shown that chemokine-induced DC migration is Ca^2+^-dependent [118]. Ca^2+^ is involved in the regulation of chemokine receptor expression, cell swelling, cytoskeletal changes and amphipod formation activities; DC migration relies tightly on the cytosolic Ca^2+^ concentration [119]. Activated DCs rapidly up-regulate chemokine receptor 7 (CCR7) expression, and acquire the ability to migrate into afferent lymphatics and drain lymph nodes [120]. CCR7 is a G protein-coupled receptor [121] that regulates DC chemotaxis, survival, migration velocity, cellularity and endocytosis; furthermore, its activation is accomplished by inducing the mobilization of intracellular calcium stores through the inositol 1,4,5-triphosphate (IP3) pathway [122,123]. Ca^2+^ plays an important role in the inflammatory response because it regulates the function of DCs in various links (Figure 3B).

VDAC1 can regulate mitochondrial function by interacting with a variety of organelles with Ca^2+^ channels, cytoplasmic proteins and OMM proteins. VDAC1 promotes efficient Ca^2+^ transfer to mitochondria by forming multiprotein complexes with Ca^2+^ channels in other organelles, such as the IP3R-VDAC1-GRP75-DJ-1 complex on the ER, RyR2-VDAC1 on the sarcoplasmic reticulum and TRPML1-VDAC1 on the lysosome [124]. VDAC1 can not only transport solutes up to 5 kDa into mitochondria, it can also mediate the transportation of various substances, including Ca^2+^, nucleotides and metabolites (pyruvate, malate, succinate, NADH/NAD, heme and cholesterol) [76,125,126]. In addition, VDAC1 can also mediate the transport of Ca^2+^ into the mitochondrial IMS through the OMM, and can also promote its transport from the IMS into the cytoplasm. Various functional properties are indirectly involved in the inflammatory response, so it is speculated that VDAC1 can become a potential target for the treatment of inflammation.

## 4. Inflammatory Diseases and VDAC1 in Energy Metabolism

Evidence suggests that altered cellular metabolism exacerbates and determines the inflammatory state of cells. Cellular metabolism constitutes a complex network of thousands of different metabolites and enzymes that are necessary for the production of nucleic acids, proteins, lipids, carbohydrates, as well as cellular energy [127,128,129]. Metabolism plays a key role in maintaining homeostasis, proliferation and cellular activation. Studies have shown that the cellular function of generating energy in the form of ATP is critical for both resting and activated cells, and is governed by tight coordination of the integrated metabolic pathway of glycolysis, the tricarboxylic acid (TCA) cycle and the pentose-phosphate pathway (PPP) [129,130,131].

VDAC1 regulates metabolites and molecular transportation. Metabolites and ions pass through the OMM via VDAC1 into the mitochondrial matrix, or are released into the cytoplasm [126]. VDAC1 affects metabolism by mediating the transport of metabolites such as pyruvate, propionate, succinate, adenosine triphosphate (ATP) and adenosine diphosphate (ADP), as well as nicotinamide adenine dinucleotide hydrogen (NAD^+^/NADH), across the mitochondria. ATP and NADP are mainly exported into the cytoplasm [126]. VDAC1 also regulates the TCA cycle by affecting intramitochondrial Ca^2+^ [132,133,134]. At the same time, Ca^2+^ can also affect the activity of mitochondrial enzymes that are located on the outer surface of the IMM, such as glycerophosphate dehydrogenase, by activating the aspartate carrier to influence the malate-aspartate shuttle as well as glutamate/malate-dependent respiration [135,136,137] (Figure 3A). Studies have shown that shutting down [138] or down-regulating VDAC1 expression reduces the exchange of metabolites between the mitochondria and the rest of the cell; this results in inhibited cell growth [76,97]. Since VDAC1 is a key protein on the outer mitochondrial membrane that contributes to metabolite and ion transportation, it is better to know what could happen if it was knocked out. Microarray analysis of VDAC1-null strain indicated that the expression of mitochondrial genes was completely reprogrammed; this was accompanied by a significant decrease in mtDNA. In order to survive, the mitochondrial metabolism became completely re-arranged, as the TCA cycle turned on the backup pattern to overcome this dysfunction [139]. Notably, VDAC1 inhibitors such as VBIT-4 did not detect toxicities, in vitro and in vivo [70], suggesting that it could be promoted to clinical trials for further investigation. Today, increasing evidence supports the immunomodulatory properties of metabolites released from glycolysis and the TCA cycle during inflammation [140]. Phosphoenolpyruvate (PEP), lactic acid, succinic acid, citric acid, etc., that are formed during metabolism have been shown to affect the inflammatory state of cells [140,141,142].

### 4.1. TCA Cycle

Metabolites in the process of energy metabolism can participate in inflammatory responses through different pathways, affecting the secretion of cytokines, the production of pro-inflammatory mediators, and the activation and differentiation of immune cells. VDAC1 plays an important role in energy metabolism, and participates in the inflammatory response by directly mediating the transport of metabolites during respiration and regulating Ca^2+^ as well as the activity of respiration-related enzymes (Figure 3A,C). Succinic acid is one of the metabolites that accumulates from the disturbance of the TCA cycle and the breakdown of hyperglutamine. Succinate accumulation leads to macrophage M1 polarization through the direct inhibition of proline hydrolase, prompting HIF-1α and IL-1β secretion [143,144]; it acts as an inflammatory stimulator in an autocrine-dependent manner [143,145]. Lipopolysaccharide (LPS)-induced succinate promotes IL-1β expression via HIF-1α signaling [144,146]. Extracellular succinate induces a pro-inflammatory response in diverse immune cells, increasing the migration and secretion of pro-inflammatory cytokines TNF-α and IL-1β in dendritic cells and macrophages [144].

Citric acid accumulates in LPS-stimulated macrophages [144,147,148]; autocrine type I IFN-driven IL-10 suppresses the activity of isocitrate dehydrogenase (IDH) and LPS-treated macrophages to promote this process [147]. Citrate is generated during the tricarboxylic acid reaction and, once in the cytoplasm, it is metabolized by ATP-citrate lyase (ACLY) to acetyl-CoA and oxaloacetate, which are precursors for lipid synthesis, ROS and NO [149]. Citrate affects ICAM-1 and cytokine (e.g., IL-6), contributing to the regulation of endothelial inflammation [150]; it acts as an anti-inflammatory factor [151,152]. Studies have suggested that reduced cytoplasmic citrate levels, due to a depletion in circulating immune complexes (CICs), reduces ROS, NO and prostaglandin production. These changes may impair the pro-inflammatory differentiation of cells, underscoring the role of certain metabolites in the inflammatory response [140,149].

### 4.2. Glycolysis

The glycolytic pathway is critical for the activation, differentiation and function of immune cells (Figure 3C). Canonically activated macrophages display pro-inflammatory properties that are primarily driven by glycolysis [153]. The metabolic switch in macrophages is controlled by glycerol-3-phosphate dehydrogenase 2 (GPD2), a key component of the glycerol phosphate shuttle that mediates the transport of electrons to mitochondria [140,154]. Phosphoglycerate dehydrogenase, the rate-limiting enzyme for de novo serine biosynthesis of glucose, is also required for macrophage M2 polarization, and is critical for macrophage function [155]. The metabolic response of T cells is similar to that of macrophages; upon activation, effectors CD4^+^ and CD8^+^ T cells shift their metabolic program toward glycolysis for faster production of ATP to meet energy demands. Studies have shown that inhibition of glycolysis with 2-deoxyglucose (2-DG) impairs the differentiation of T helper type 1 (Th1) and Th2 cells [156]. In addition, glycolytic enzymes can act as post-transcriptional regulators of inflammatory genes, and the classical glycolytic enzyme GAPDH can bind to mRNA. In CD4^+^ T cells, GAPDH inhibits their translation by binding to IFN-γ, c-Myc, granulocyte macrophage colony-stimulating factor (GM-CSF) and IL-2 mRNA at the AU-rich region [140,157,158].

Glycolysis is essential for the maturation and function of both cDCs and GM-DCs (GM-CSF-induced DCs); it is critical for DC activation. With 2-DG, a glycolytic inhibitor that inhibits hexokinase (HK), treatment impairs GM-DC and conventional DCs costimulatory marker expression and IL-12 production, as well as their primary function on T cells [159,160]. Glycolytic activity is critical for DC migration; glucose-deprived GM-DCs show decreased mobility, loss of rounded morphology, increased dendrites and impaired oligomerization of CCR7, a chemotactic driver that drives DC migration to lymph nodes factor receptors [161]. The glycogen phosphorylase inhibitor CP91149 disrupts glycogen metabolism and significantly impairs conventional DC maturation and function, especially at the earliest stages of GM-DC activation. Disruption of the glucose-pyruvate pathway significantly impairs DC maturation, costimulatory molecule up-regulation, cytokine secretion and T-cell stimulatory capacity [161,162].

Phosphoenolpyruvate (PEP) is produced by enolase-1 during glycolysis, and accumulates in T cells. The accumulation of PEP has a similar pro-inflammatory effect on macrophages, promoting M1 poles, which increases the expression of pro-inflammatory cytokines [140,163]. PEP is associated with inflammation via its impact on Ca^2+^ [140]. PEP can inhibit the ER calcium channel to suppress Ca^2+^ flux to the ER [140], resulting in the increased cytoplasmic Ca^2+^ promoting the activation of nuclear factor of activated T cells [140,164]. Lactic acid, the final product of glycolysis, can display signaling properties during inflammation [165]. During this process, lactate suppresses immune responses by impairing the shift in metabolic reorganization to a pro-inflammatory phenotype and blocking pro-inflammatory signaling pathways in monocytes, macrophages and DCs [166,167]. The accumulation of lactate in DCs drives the switch to an anti-inflammatory phenotype by increasing IL-10 [168]. However, lactate-rich environments have been reported to enhance Th17 responses in macrophages [169]. Lactate can promote Th17 responses and activate NF-κB pro-inflammatory signaling of macrophages [169,170]. Lactate is able to enter cells, stimulate the NF-κB/IL-8 pathway and induce ROS production [171]. Lactate plays a key role in the regulating macrophage polarization, modification of histones and the inflammatory response [172,173]; it also enhances IFN-γ expression and the differentiation of T helper 1 cell [174]. The various roles of lactate in inflammatory processes have been recently documented [175]. 

The initial and rate-limiting steps of glycolysis are mainly catalyzed by HK1, most of which is bound to the OMM, mainly through mitochondria formed by VDAC1 and adenine nucleotide translocator (ANT) intermembrane contact sites for transport [176]. It has also been shown that Hexokinase-2 (HK2) binds to VDAC1 on the OMM to facilitate the preferential entry of ATP into HK2 for glycolysis [177]. The binding of HK2 with mitochondrial VDAC1 can be inhibited by chrysin, resulting in decreased glucose uptake and lactate production [178]. VDAC1 is directly involved in the regulation of the glycolytic pathway; it affects the activation, differentiation and migration of various immune cells such as macrophages, DCs, T cells, etc., and affects the production, migration, and release of various cytokines and pro-inflammatory mediators. 

VDAC1 can affect mitochondrial respiration, as a result of its important role in controlling the transportation of substances and metabolites. The intermediates in the Krebs cycle have a close relation with the inflammation process [127]. The metabolism of PEP, lactic acid, succinic acid, citric acid, etc., plays an important role in the occurrence and development of inflammation. In conclusion, VDAC1 could become a new therapeutic target for inflammation, and this necessitates further study.

## 5. Inflammatory Diseases and VDAC1 in Lipid Metabolism

Lipid metabolism is an important and complex biochemical reaction in the body; it is the process of digestion, absorption and decomposition of fat in the body through the help of various related enzymes, and is of great significance to vital activities [179,180,181,182]. Diseases caused by abnormal fat metabolism have become common, such as non-alcoholic steatohepatitis (NASH), hyperlipidemia, cardiovascular and cerebrovascular diseases, etc.

VDAC1 is involved in cholesterol transport, and is generally considered to be part of a complex that mediates fatty acid transport through the OMM [75,125,183]. Meanwhile, VDAC1 also serves as an anchoring site for long-chain acyl-CoA synthase (ACSL), which is associated with the outer surface of the OMM, and for carnitine palmitoyltransferase 1a (CPT1a), which faces the intermembrane space [183] (Figure 3A). ACSL catalyzes the synthesis of fatty acyl-CoA in vivo, which is the first reaction in the human body to utilize fatty acids; meanwhile, CPT1a is involved in the process of transporting long-chain fatty acids into the mitochondria so that fatty acids can be broken down to generate usable energy for cells. It has been reported that CPT1a, ACSL and VDAC1 can form a complex, and that the long-chain fatty acyl-CoA synthesized by ACSL is transferred from the OMM to the intermembrane space through VDAC1; furthermore, CPT1a converts acyl-CoA into long-chain fatty acylcarnitine [183], followed by a series of subsequent oxidation reactions.

It has been found that the phosphorylation state of VDAC1 mediated by glycogen synthase kinase 3 (GSK3) can control the permeability of the OMM [184]. It has been observed that a loss of VDAC1 may cause mitochondria to stop oxidizing fatty acids, and VDAC1 inhibitors can inhibit palmitate oxidation [185,186]. In addition, the VDAC1-based peptide, R-Tf-D-LP4, can stimulate catabolic pathways that are involved in promoting fatty acid transfer to the mitochondria, fatty acid oxidation and increasing the expressions of enzymes and factors that are associated with fatty acid transport to the mitochondria, thereby enhancing β-oxidation and production of energy [185]. There are experimental results that show that R-Tf-D-LP4 significantly reduces pathophysiological features, such as hepatocyte ballooning, and inflammation and liver fibrosis in the HFD-32/STAM mouse model that is associated with steatohepatitis and/or NASH; meanwhile, this peptide also reduces the expression of inflammatory macrophages and cytokines (IL-1β and IL-6) in the liver of HFD-32-fed mice [185]. Dysfunction or deletion of VDAC1 will lead to fat deposition and abnormal lipid metabolism, increasing inflammatory macrophages and the expression of cytokines (IL-1β and IL-6). Heightened expression of IL-6 increases autocrine IL-4, which enhances Th2-type immune responses through automated feedback loops, playing an important role in inflammation [185]. 

These findings indicate that VDAC1, as a key factor in mitochondrial lipid metabolism, can regulate the oxidative decomposition of fat. Abnormal mitochondrial lipid metabolism caused by its dysfunction will lead to the blocking of oxidative reaction, abnormal modification and localization of lipoproteins, etc., which may be related to subsequent causes of the inflammatory response, and to inflammation-related diseases. 

## 6. Inflammatory Diseases Pathogenesis and VDAC1 in Mitophagy

Mitophagy maintains the functional integrity of the mitochondrial network and cellular homeostasis by selectively sequestering and degrading damaged or incomplete mitochondria [187]. As previously described, VDAC1 is associated with inflammation via various signaling pathways. However, little research has focused on whether and how VDAC1 is involved in inflammatory diseases pathogenesis via mitophagy. In this section, we discuss two major questions. We hope this will lead to more investigations into how VDAC1 contributes to inflammatory diseases pathogenesis via mitophagy. (1) What potential signaling does VDAC1 use to regulate inflammation via mitophagy? (2) VDAC1 controls mtDNA release and promotes inflammation; how does mitophagy modulate mtDNA levels?

### 6.1. Mitophagy Regulates Inflammation via VDAC1

Mitophagy plays a key role in the regulation of inflammatory signaling, and the process of mitophagy limits inflammatory cytokines secretion [36,37,188,189,190] (Figure 4). 

The PTEN-induced putative kinase 1 (PINK1) and the RING family ubiquitin ligase Parkin were found to be involved in mitophagy [191,192,193,194]. This indicates that induced mitophagy can be accomplished in cells that overexpress Parkin or overexpress PINK1. PINK1/Parkin acts as key regulator of mitophagy, and is vital in controlling infection and the inflammatory response [195]. The interaction of two Parkin domains, RING1 and ubiquitin-like (UBL), affects its activity. UBL binding with RING1 results in the inactive state of Parkin; PINK1 phosphorylates UBL-Ser65, leading to the activation of Parkin to promote substrate ubiquitination, with VDAC1 included [196]. 

Studies have demonstrated that Parkin interacts with VDACs, and that VDAC1 is the target of Parkin-mediated Lys27 polyubiquitination and mitochondrial phagocytosis [27,197,198,199]. VDACs are effective in helping Parkin identify defective mitochondria, and assist in the subsequent mitochondrial phagocytosis. VDAC1 is necessary for PINK1/Parkin to target damaged mitochondria [197]. Partial silencing of VDAC1 resulted in significantly reduced Parkin translocation from the cytoplasm to damaged mitochondria, while also significantly preventing mitochondrial clearance. Notably, retransfection of flag-tagged VDAC1 significantly restored Parkin mitochondrial translocation and clearance [197]. Parkin ubiquitinates VDAC1, and ultimately selectively degrades damaged mitochondria by promoting mitophagy [199]. Notably, the ubiquitylation of VDAC1 was observed with enhanced expression of Parkin instead of endogenous Parkin [197]. It has been shown that Parkin’s targeting of defective mitochondria is impaired in the absence of both VDACs, but that it can be rescued by expressing VDAC1 or VDAC3 in these cells [199]. These pieces of evidence confirm that VDAC1 is important for PINK1/Parkin-involved mitophagy (Figure 4A). VDAC1 channels and the PINK1 pathway are closely related to impaired mitophagy-associated inflammation. Mitophagic stimulation could reverse memory impairment via PINK1 signaling in Alzheimer’s disease models [200]. 

The anti-inflammatory mechanism of mitophagy may be achieved by inhibiting the excess production of IL-1β and IL-18 [201]. Viruses can exploit the inhibitory effect of mitophagy on IL-1β and IL-18 secretion to evade pathogen clearance [202]. Type I IFNs are a group of pleiotropic cytokines, including IFN-α and IFN-β, that promote antigen presentation, NK cell function and lymphocytic responses [203]. Mitophagy could inhibit type I IFN synthesis [204] (Figure 4B). Studies have shown that viruses can utilize a unique mitophagic pathway to attenuate type I IFN responses to viral replication [205,206,207]. Mitochondrial antiviral signaling (MAVS) plays an important role in the regulation of mitochondrial homeostasis and the native immune response. Mitophagy controls MAVS that mediates antiviral signaling [208]. MAVS activates NF-κB and IRF3 signaling in response to viral infection, resulting in enhanced type I IFN levels [208,209]. Immunity related GTPase M (IRGM), an autoimmunity gene, contributes to regulating the interferon response by attenuating cGAS-STING and RIG-I-MAVS signaling [210]. Notably, knockout of IRGM results in mitophagic deficiency as well [210]. Viral glycoprotein can induce mitophagy, causing inhibition of the IFN response via promoting MAVS degradation [211]. VDAC1 is one of the vital IL-1β regulatory genes [212]. The hypo-methylation of VDAC1 promoter leads to enhanced VDAC1 levels, resulting in overexpression of IL-1β [212]. Inhibition of VDAC1 leads to an attenuation in TNF-α induced VCAM-1 expression [213]. Attenuation in the activity of the VDAC1 channel would suppress IL-33 release [214]. IL-33 promotes ROS levels, and effects mitophagy by activating AMPK signaling [215]. The oligomerization of VDAC1 is associated with the interaction with mtDNA [32,33]. Released mtDNA could trigger a type I IFN response [32]; inhibition of VDAC1 oligomerization results in decreased mtDNA release and type I IFN signaling [32].

Mitophagy has been shown to have anti-inflammatory effects by down-regulating inflammasomes [216]. Increasing evidence suggests that the inhibitory role of mitophagy in NLRP3 inflammasome activation is attributable to impaired mitophagy [88,217]; previous controlled experiments found increased IL-1β secretion in autophagic mitochondria-deficient macrophages, accompanied by accumulations of damaged mitochondria, upon stimulation of various NLRP3 activators [88]. A recent study found that mitophagy can directly target NLRP3 inflammasome components and IL-1β for lysosomal degradation [218]. The activation of NLRP3 inflammasomes and type I IFN signaling are highly associated with VDAC1 channels [33]. A natural product, oleanolic acid, shows anti-inflammatory effects by suppressing NLRP3 inflammasomes; it does so by decreasing VDAC1 expression and stimulating the overproduction of oxygen species [219]. 

Taken together, VDAC1 could be in the principal position to affect the mitophagic response to inflammation that results in cytokines release, IL-1β, IL-18, IFN-α, IFN-β, etc. VDAC1 is the hub of mitophagy, inflammasomes and inflammatory immune responses.

### 6.2. Mitophagy Modulates mtDNA Levels in Cytoplasm

The mtDNA released into the cytoplasm may lead to the occurrence of inflammatory responses through the activation of TLR9, NLRP3 inflammasomes, AIM2 inflammasomes and the cGAS-STING pathway [65,220], and promoting IL-1β production [221]. mtDNA also mediates inflammatory responses by activating IL-1 receptors, promoting neutrophil migration and macrophage responses, promoting T cell differentiation and function, as well as NK cell recruitment [188,222,223,224]. Mitophagy is the key mechanism that inhibits mtDNA release [225]. Defective mitophagy enhances cytoplasmic mtDNA levels. Aging results from mitochondrial injury and impaired mitophagic activation in macrophages. Increased mtDNA in the cytoplasm promotes STING activation of aged macrophages. The mitophagy mediated mtDNA-cGAS-STING pathway is involved in different sterile inflammatory responses [226] and mitochondrial diseases [227]. TNF impacts mitochondrial function, and blocks mitophagy, and results in mtDNA release and the cGAS-STING-dependent interferon responses of inflammatory arthritis [228]. The mtDNA is a consequence of the impaired mitophagy [34,229,230,231] of PRKN/PINK1 parkinsonism [229], which is associated with high levels of cytokine IL-6 [229,232]. In the absence of mitophagy, mtDNA release activates the NF-kB pathway via TLR9, resulting in enhanced transcription of multiple inflammatory cytokines, including TNF-α and IL-6 [220]. Hepatocyte-specific XBP1 knockout mice were found to have impaired mitophagy that resulted from increased mtDNA release via the cGAS-STING pathway of macrophages in the thioacetamide-induced acute liver injury model [233]. The same signaling mediates aging macrophages, with defective mitophagy enhancing the mtDNA cytosolic release of liver sterile inflammation [226]. Additionally, impaired mitochondrial integrity, mitophagy, results in accumulating mtDNA in the cytosol of murine cardiac anomalies models via cGAS-STING-TBK1 signaling [234]. Neutrophils release oxidized mtDNA to drive type I IFN of human systemic lupus erythematosus (SLE) [235]. Lack of proper clearance of neutrophil-released mtDNA may be the key pathogenesis of SLE [235]. VDAC1 is vital in mtDNA transportation. By targeting VDAC1-mtDNA, a novel therapeutic approach to human lupus could be uncovered.

Cytosolic mtDNA triggers inflammatory responses [236]. Mitophagy results in enhanced cytoplasmic mtDNA, which contributes to inflammation that is associated with lung injury through the TLR9-MyD88-NF-κB pathway [231,237]. These indications from targeting mitophagy-mtDNA-related signaling could provide novel and promising therapeutic strategies.

The process of mitophagy may limit the secretion of inflammatory factors that directly regulate mitochondrial antigen presentation and immune cell homeostasis [188] (Figure 4B). All the evidence indicates that mitophagy inhibits the occurrence and development of inflammatory responses by affecting the secretion of inflammatory cytokines, in addition to the maturation, differentiation and function of immune cells. The role of mitophagy in the occurrence and development of inflammation and autoimmune diseases cannot be ignored. PINK1-PRKN/PARK2-mediated mitophagy is the most extensively studied Ub-dependent pathway [188]. The interaction between VDAC1 and Parkin is important for PINK1/Parkin-directed mitophagy. VDAC1 regulates the release of inflammatory cytokines that result from mitophagy, by affecting this pathway; moreover, the activation and differentiation of immune cells play an important role in inflammation and autoimmune diseases. Recent studies found that this autophagic pathway may be closely related to many diseases, such as primary biliary cirrhosis (PBC) [238], SLE [239], asthma [240,241], including eosinophilic airway inflammation [242,243], airway hyperresponsiveness, and airway remodeling [241,244,245,246]. The regulation of this pathway involving VDAC1 for the treatment of inflammation-related diseases is of great clinical significance and deserves further exploration and research.

## 7. Summary and Conclusions, Current Clinical Conditions and Future Perspectives

Mitochondria are fundamental organelles that execute and coordinate various metabolic processes in cells. Mitochondria are key organelles that are associated with cellular functions, and well-functioning mitochondria are critical to maintaining tissue homeostasis [247]. Mitochondrial malfunction is a sign of oxidative stress, inflammation, aging and chronic degenerative diseases [247,248,249]. VDAC1 is an important regulator of mitochondrial function, and acts as a mitochondrial gatekeeper that is responsible for cell fate [126]. As a multifunctional protein channel, VDAC1 can coordinate the transport of proteins and metabolites, and regulates apoptosis as well as other cellular stress-related processes [126,250]. In addition, VDAC1 can also participate in the regulation of inflammation by affecting the respiratory chain and promoting the expression of cytokines. VDAC1 is also involved in the production and metabolism of mitochondrial energy, regulating mitochondrial lipid metabolism and regulating mitophagy, all of which indicate that it is a promising target for novel therapeutic strategies [29,126]. 

Cutting-edge research confirms that VDAC is essential for the apoptotic “Find me signaling” pathway that results from the failure of apoptotic cell clearance, and leads to the pathogenesis of cystic fibrosis, followed by sterile inflammation [251]. Ulcerative colitis (UC) may be promoted by VDAC1 overexpression, and novel interacting targets for the treatment of UC based on VDAC1 are being developed for inflammatory and/or autoimmune diseases [74]. In addition, research suggests that VDAC1 is also related to cardiovascular and cerebrovascular diseases [252]. Furthermore, VDAC1 is widely involved in cancer [28,253], neurodegenerative diseases [254,255], diabetes [72,256], kidney disease [257,258], aging [259] and other areas of study. These all suggest that VDAC1 is a reasonable target to develop the next generation of therapeutic drugs. 

In recent years, we have witnessed a considerable accumulation of knowledge about the function of VDACs. Biochemical, molecular and biophysical approaches have advanced our understanding of the structure-function relationships of VDAC, and have uncovered a diversity of regulatory mechanisms that control VDAC1 function. The high-resolution structure of recombinant VDAC1 has been determined, and VDAC1 β-strands have been identified; dimerization sites, Ile-27, Leu-29, Thr-51 and Leu-227 have been found to be involved in VDAC1 oligomerization [260]. Deep insights into mechanism regulation still necessitate further investigation.

mtDNA inflammation-related clinical trials (NCT03929458, NCT04078035 and NCT04045223) were completed, but no results were released; other trials are still ongoing (NCT03077672, NCT05441787, NCT03938909 and NCT04334499). There is only one mitophagy inflammation clinical trial (NCT05040503), which is still ongoing. VDA-1102 was designed to modulate VDAC/HK2, which effects glycolysis and mitochondrial function in cancer and activated immune cells. VDA-1102-related clinical trials have been conducted against solid tumors by VidacPharma, and no serious adverse events were found from a Phase II B (NCT 03538951) study (http://www.vidacpharma.com/clinical-trials, accessed on 20 September 2022).

In conclusion, VDAC1 is closely associated with mtDNA and cytokines release, with the former being a messenger of impaired mitophagy. Mitophagy regulates inflammation via VDAC1. VDAC1 plays a principal and pivotal role in maintaining mitochondrial homeostasis and inflammation-related immune responses. Further investigation of VDAC1 and its related pathways may provide promising therapeutic strategies against multiple inflammation-associated diseases. We highly expect VDAC1 to become a therapeutic target, and we hope our research leads to novel treatment strategies and breakthroughs for many diseases via VDAC1. 

## Figures and Tables

**Figure 1 cells-11-03174-f001:**
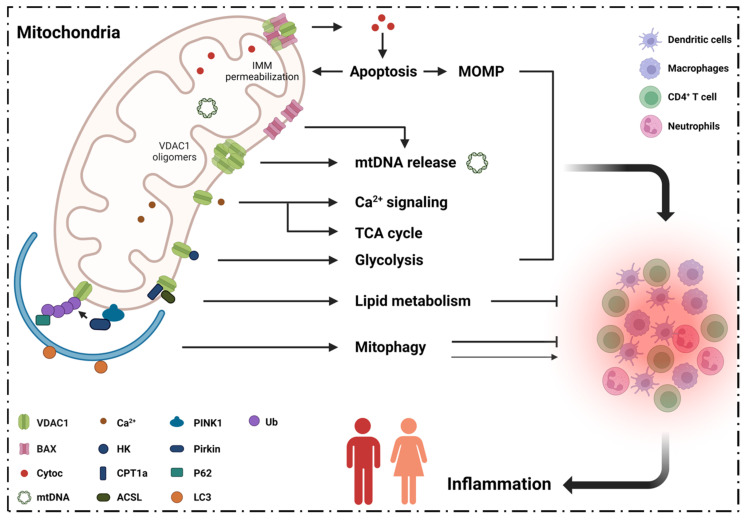
**VDAC1 regulates inflammatory pathogenesis.** Mitochondria are the center of energy generation, the TCA cycle, glycolysis and lipid metabolism. VDAC1 is the fundamental component that maintains mitochondrial function. VDAC1 plays an important role in the regulation of apoptosis, mtDNA release, Ca^2+^ signaling, TCA cycle, glycolysis, lipid metabolism and mitophagy. Impaired mitochondrial homeostasis with dysfunctional signal networks results in inflammatory pathogenesis and mitochondrial diseases. **Abbreviations:** ACSL: long-chain acyl-CoA synthase; BAX: Bcl-2-associated X protein; CPT1a: carnitine palmitoyltransferase 1A; Cyto c: cytochrome c; HK: hexokinase; LC3: microtubule-associated proteins 1A/1B light chain 3; IMM: inner mitochondrial membrane; MOMP: mitochondrial outer membrane permeabilization; mtDNA: mitochondrial DNA; PINK1: PTEN-induced putative kinase 1; TCA cycle: tricarboxylic acid cycle; Ub: ubiquitin; VDAC1: voltage-dependent anion channel 1.

**Figure 2 cells-11-03174-f002:**
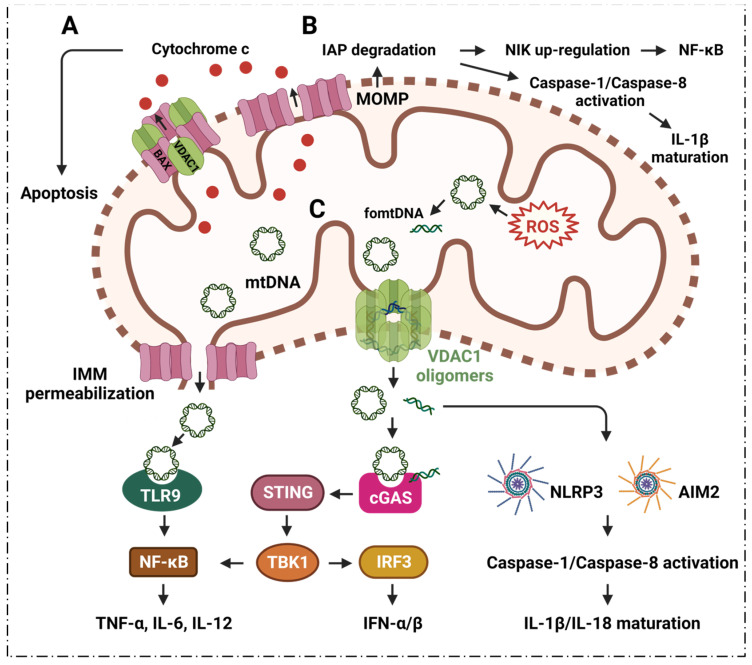
**VDAC1 mediates apoptosis and mtDNA release to promote cytokines expression and inflammatory pathogenesis.** (**A**) Bcl-2 family member, BAX, interacts with VDAC1 to release cytochrome c into the cytoplasm, promoting apoptosis. (**B**) MOMP induces proteasomal degradation of IAPs, which causes NIK to further induce the pro-inflammatory NF-κB signal and activate caspase-1/8; this in turn results in the maturation of pro-inflammatory factor IL-1β. (**C**) Mitochondrial overproduced ROS oxidize mtDNA to fomtDNA. The mtDNA and fomtDNA pass the VDAC1 oligomers channel or the oligomerization BAX pore into the cytoplasm. The released mtDNA/fomtDNA induce the cGAS-STING pathway to promote interferon gene expressions via TBK1-IRF3 to up-regulate IFN-α/β, or through TBK1-NF-κB to enhance TNF-α, IL-6 and IL-12. Additionally, mtDNA interacts with TLR9 and promotes TNF-α, IL-6 and IL-12 expression via NF-κB signaling. Moreover, the released mtDNA induces the NLRP3 inflammasome and AIM2 inflammasome to enhance caspase-1/8 activation to promote IL-1β/IL-18 maturation. **Abbreviations:** AIM2: absent in melanoma 2; BAX: Bcl-2-associated X protein; cGAS: cyclic GMP-AMP synthase; Cyto c: cytochrome c; fomtDNA: oxidized mtDNA fragments; IAP: inhibitors of apoptosis; IFN: interferon; IL: interleukin; IRF3: interferon regulatory factor 3; IMM: inner mitochondrial membrane; MOMP: mitochondrial outer membrane permeabilization; mtDNA: mitochondrial DNA; NF-κB: nuclear factor-κB; NIK: NF-κB induced kinase; NLRP3: nucleotide-binding domain and leucine-rich repeat (LRR) containing P3; ROS: reactive oxygen species; STING: stimulator of interferon genes; TLR9: Toll-like receptor 9; TBK1: TANK-binding kinase 1; TNF: tumor necrosis factor; VDAC1: voltage-dependent anion channel 1.

**Figure 3 cells-11-03174-f003:**
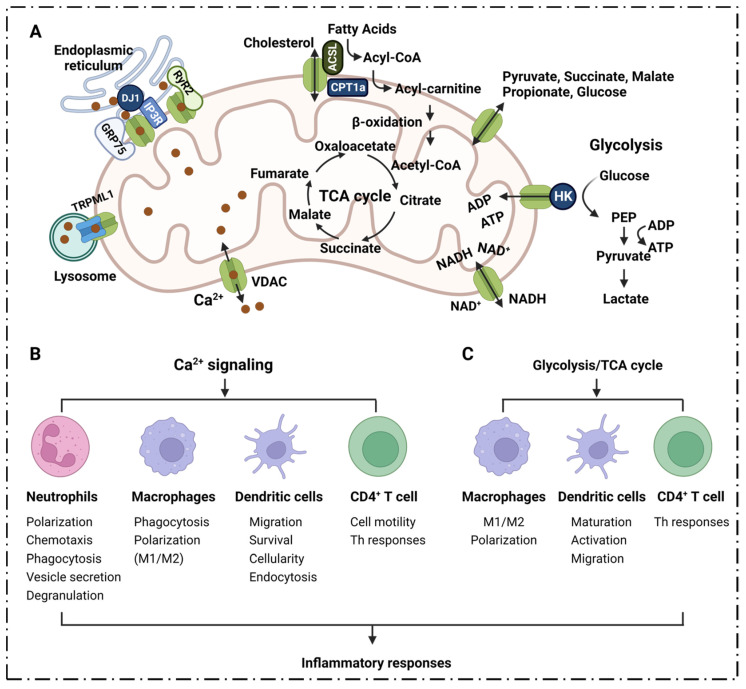
**VDAC1 is involved in Ca^2+^ transportation, lipid metabolism, glycolysis****, TCA cycle****and in inflammatory responses.** (**A**) VDAC1 regulates Ca^2+^ transportation, glycolysis, TCA cycle and lipid metabolism. VDAC1 transports Ca^2+^ between the mitochondria and cytoplasm to maintain calcium homeostasis. In the energy generation system, the VDAC1 pore maintains substrates, metabolites, biomolecules, etc., in a balanced manner to sustain salutogenesis. (**B**) Ca^2+^ signaling affects the inflammatory responses of neutrophils, macrophages, dendritic cells and CD4^+^ T cells. (**C**) Inflammatory responses of macrophages, dendritic cells and CD4^+^ T cells are promoted by glycolysis/TCA cycle energy generation pathways. **Abbreviations:** VDAC1: voltage-dependent anion channel 1; TRPML1: also known as MCOLN1, mucolipin TRP cation channel 1; GRP75: glucose-regulated protein 75; IP3R: inositol 1,4,5-trisphosphate receptor; DJ1: deglycase DJ-1, also known as Parkinson disease protein 7, is encoded by the PARK7 gene in human; RyR2: ryanodine receptor 2; CPT1a: carnitine palmitoyltransferase 1A; ACSL: long-chain acyl-CoA synthase; TCA cycle: tricarboxylic acid cycle; HK: hexokinase; ATP: adenosine triphosphate; ADP: adenosine diphosphate; NADH: nicotinamide adenine dinucleotide hydrogen; PEP: phosphoenolpyruvate; Th: T helper.

**Figure 4 cells-11-03174-f004:**
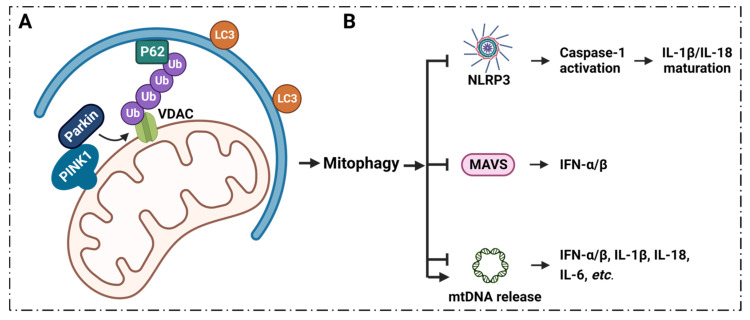
**VDAC1, PINK1/Parkin signaling and mitophagy.** (**A**) PINK1/Parkin targets damaged mitochondria, ubiquitinates VDAC1, and ultimately degrades damaged mitochondria by promoting mitophagy. (**B**) Mitophagy modulates NLRP3, MAVS and mtDNA release, affecting the immune response. NLRP3 activates caspase-1 to promote IL-1β/IL-18 maturation. MAVS enhance IFN-α/β expression. Meanwhile, mitophagy suppresses NLRP3, MAVS and mtDNA release, which results in reduced cytokines release. Additionally, mitophagy can also promote mtDNA release, which affects cytokines expression. **Abbreviations:** IL: interleukin; IFN: interferon; LC3: microtubule-associated proteins 1A/1B light chain 3; MAVS: mitochondrial antiviral signaling protein; NLRP3: nucleotide-binding domain and leucine-rich repeat (LRR) containing P3; PINK1: PTEN-induced putative kinase 1;; Ub: ubiquitin; VDAC1: voltage-dependent anion channel 1.

## Data Availability

Not applicable.

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
