# Peer review of "Mitochondrial VDAC1: A Potential Therapeutic Target of Inflammation-Related Diseases and Clinical Opportunities"

_cells, 2022, doi:10.3390/cells11193174_

Round 1
Reviewer 1 Report
The review by Hu and colleagues focus on VDAC1 involvement in inflammation diseases, a field that was less explored in comparison to others, such as cancer and neurodegeneration. In this perspective, the review could provide a different and partial inedited point of view to researchers working in the inflammation field. However, before the manuscript is taken in consideration for the publication, there are several concerns to fulfill about the general setting of the review, the reference list and the language.
The review is focused on VDAC1 albeit the information about the protein are distributed along the whole manuscript and are often incomplete. In my opinion, an exhaustive and introductive paragraph describing the molecular structural and functional features of the protein is mandatory. Specifically, authors should dissect the main VDAC aspects and cite the correct papers. Precisely, description of 3D structure, as in lane 98-102, should be moved in the introductive paragraph and described in-depth; in this case, there are three papers by independent groups that have discovered the structure and these references must be included (here the doi: 10.1073/pnas.0808115105; 10.1126/science.1161302; 10.1073/pnas.0809634105). Other important aspect to describe in this introductive paragraph are:
- the voltage-dependent features discovered in 1976 (10.1007/BF01869662) and confirmed by recent works (as the following: 10.1016/j.bbabio.2018.01.008);
- the evolution of VDAC proteins (see as example 10.3389/fphys.2021.675708) and the emerging isoform-specific functions, as for example the anti-apoptotic functions of VDAC2 (10.1126/science.1083995) and the recent involvement of VDAC3 in the redox metabolism (10.1016/j.redox.2022.102264);
- the regulation of gene(s) expression as recently determined (10.3390/ijms21197388; 10.3389/fphys.2021.708695).
Paragraph 6 appears too long and partially lose the focus of the review. If 6.1 connects VDAC1 to the regulation of mitophagy, paragraph 6.2 is mostly irrelevant for the review. I had a similar feeling for paragraph 3. I strongly recommend to revise the paragraphs, by reducing the amount of information that are not directly related to VDAC and, therefore, to the aim of the review.
Sometimes, authors use imprecise words or sentences. For instance, what they mean with “vital” in lane 39? I guess all organelles or cellular components can be considered “vital”. In lane 40, what are the “normal cell biology activities”? Please, revise these sentences.
Several specific references are missing. For example, in lane 64 authors describe the involvement of VDAC1 in cancer and neurodegeneration as well as the use of VDAC as a therapeutic target. Authors should briefly describe it in detail and cite the appropriate reference. Also, what is the reference for the sentence in lane 117-119?
Lane 94: I’m not sure that MOM can be considered a part of the MPTP; it is true that VDAC1 is located in the MOM, but the other component, ANT, is in the inner membrane. Is this mean that also the IMM belongs to the MPTP? Please revise the sentence.
Lane 341: the passage of metabolites across VDAC1 is bi-directional. Authors should clearly state that while ADP and NAD+ are normally imported into mitochondria, ATP and NADH are mainly exported. Thus, using the word “into” is not correct.
Lane 346: authors haven’t take in consideration a relative work recently published by De Pinto group in which it has been demonstrated that VDAC1 knock-out not only decrease metabolites exchanges across the mitochondria, but also it promote a complete rewiring of the whole cell metabolism (doi: 10.1007/s00018-019-03342-8). Authors should add these information and cite the appropriate reference.
I strongly recommend an overall revision of the language and the grammar. Following several examples:
Lane 34: “in moderation” should be replaced with “when moderate”.
Lane 35: “that resulting” should be “that results” or “resulting”.
Lane 66: the semicolon should be removed.
Lane 85: “It also indicated” should be “It has been also indicated”.
Lane 117: MOMP is repeated twice; the second should be replaced with “that”.
Lane 133: replace “and” with a comma.
Lane 274: the sentence “…in phagocytosis, elevated cytosolic calcium…” should be replaced with “…in phagocytosis; indeed, elevated cytosolic calcium…”
Lane 290: remove “et al”
In many cases, the comma before “and” should be removed.
Author Response
Reviewer 1
The review by Hu and colleagues focus on VDAC1 involvement in inflammation diseases, a field that was less explored in comparison to others, such as cancer and neurodegeneration. In this perspective, the review could provide a different and partial inedited point of view to researchers working in the inflammation field. However, before the manuscript is taken in consideration for the publication, there are several concerns to fulfill about the general setting of the review, the reference list and the language.
Response: Thank you for your review of our manuscript. The authors appreciate the constructive comments that have greatly helped us improve our manuscript. We have seriously revised the manuscript, please kindly check the tracking version and the clean version marked in red. The following are the point by point response.
Reviewer 1 Comment 1
The review is focused on VDAC1 albeit the information about the protein are distributed along the whole manuscript and are often incomplete. In my opinion, an exhaustive and introductive paragraph describing the molecular structural and functional features of the protein is mandatory. Specifically, authors should dissect the main VDAC aspects and cite the correct papers. Precisely, description of 3D structure, as in lane 98-102, should be moved in the introductive paragraph and described in-depth; in this case, there are three papers by independent groups that have discovered the structure and these references must be included (here the doi: 10.1073/pnas.0808115105; 10.1126/science.1161302; 10.1073/pnas.0809634105). Other important aspect to describe in this introductive paragraph are: The voltage-dependent features discovered in 1976 (10.1007/BF01869662) and confirmed by recent works (as the following: 10.1016/j.bbabio.2018.01.008);
Response: Thank you for the valuable and constructive comment. The manuscript had been seriously revised.
The discovery of VDAC description in the introduction:
“Voltage dependent anion-selective channel was obtained from paramecium mitochondria in 1976 [1]. Now we know there are two isoforms of voltage-dependent anion channel (VDAC) in yeast, yVDAC1 and yVDAC2, with yVDAC1 is the most abundant [2, 3]. Three VDAC family members in mammalian mitochondria were observed, VDAC1, VDAC2, VDAC3; with VDAC1 being the most widely expressed and contributing to a broad and general role [4-6].” In the revised clean version lane 46-51.
The description of 3D structure in initial version lane 98-102 had been moved to the introduction part, “The 3-dimensional structure of VDAC1 shows that the 19 transmembrane β-strands form a flexible loop, forming a β-pore containing a 25-residue in the N-terminal domain. The migration of the N-terminal domain is involved in channel gating and the formation of VDAC1 dimers that transports metabolites and molecules to maintain mitochondria homeostasis [4, 7, 8].” In the revised clean version lane 62-66.
The additional VDAC1 structure and related description in the introduction:
“This VDAC1 structure was released by three independent groups in 2008 [9-11]. The β1 (26Leu-Ile-Lys-Leu-Asp-Leu-Lys-Thr-Lys-Ser35) and β19 (273His-Lys-Leu-Gly-Leu-Gly-Leu-Glu-Phe-Gln282) strands are parallel [9, 10]. Bcl-2 protein Bcl-xL interacts with β17 (243Ile-Gly-Leu-Gly-Tyr-Thr-Gln-Thr-Leu251) and β18 (255Ile-Lys-Leu-Thr-Leu-Ser-Ala-Leu-Leu263) fulfilling an anti-apoptotic function by suppressing mitochondrial release apoptogenic proteins [10]. The conserved and flexible sequence (21Gly-Tyr-Gly-Phe-Gly25) acts as a bridge connecting α-helix to β1 [11]. The α-helix is in the midway of the barrel pore at a horizon position [9]; it acts as a gate by narrowing the pore cavity to modulate the metabolites transportation [11].” In the revised clean version lane 66-74.
Reviewer 1 Comment 2
The evolution of VDAC proteins (see as example 10.3389/fphys.2021.675708) and the emerging isoform-specific functions, as for example the anti-apoptotic functions of VDAC2 (10.1126/science.1083995) and the recent involvement of VDAC3 in the redox metabolism (10.1016/j.redox.2022.102264); The regulation of gene(s) expression as recently determined (10.3390/ijms21197388; 10.3389/fphys.2021.708695).
Response: Thank you for the comment. The manuscript had been revised.
“Now we know there are two isoforms of voltage-dependent anion channel (VDAC) in yeast, yVDAC1 and yVDAC2, with yVDAC1 is the most abundant [2, 3]. Three VDAC family members in mammalian mitochondria were observed, VDAC1, VDAC2, VDAC3; with VDAC1 being the most widely expressed and contributing to a broad and general role [4-6]. Notably, VDAC2 in mammals, contributes to anti-apoptotic phenotype by binding to Bcl-2 homologous antagonist killer (BAK); the mitochondrial apoptosis is activated resulting from the homo-oligomerization of BAK when VDAC2 is displaced by truncated BH3 interacting-domain death agonist (tBID), Bcl-2-like protein 11 (BIM) or Bcl-2 associated agonist of cell death (BAD) [12]. VDAC3, especially the indispensable cysteine residues, plays an important role to protect mitochondria from oxidative stress [13]. The transcriptional factors that regulate cell growth, apoptosis, energy metabolism, etc. are also regulating VDAC genes expression [14]. More information on VDAC isoforms gene regulation had been documented by Vito De Pinto et al. [6, 14]. In this review, we will only focus on the most abundant isoform VDAC1 and its relationship with inflammation.” In the revised clean version lane 47-61.
Reviewer 1 Comment 3
Paragraph 6 appears too long and partially lose the focus of the review. If 6.1 connects VDAC1 to the regulation of mitophagy, paragraph 6.2 is mostly irrelevant for the review. I had a similar feeling for paragraph 3. I strongly recommend to revise the paragraphs, by reducing the amount of information that are not directly related to VDAC and, therefore, to the aim of the review.
Response: Thank you for the constructive comment. We have seriously revised the section 6 in the manuscript. The Figure 4 has been modified accordingly.
We are focusing on two major question in this part “1) The potential signaling that VDAC1 regulates inflammation via mitophagy. 2) VDAC1 controls mtDNA release and promotes inflammtion. How mitophagy modulates mtDNA levels?” We hope this will lead to more investigations of how VDAC1 contributes to inflammation diseases pathogenesis via mitophagy. In the revised clean version lane 532-657.
Reviewer 1 Comment 4
Sometimes, authors use imprecise words or sentences. For instance, what they mean with “vital” in lane 39? I guess all organelles or cellular components can be considered “vital”. In lane 40, what are the “normal cell biology activities”? Please, revise these sentences.
Response: Thank you for the comment. Yes, all the organelles or cellular components can be considered “vital”. We revised the sentences accordingly throughout the manuscript.
Original lane 39 and lane 40: “Mitochondria are one of the vital organelles in eukaryotic cells that are not only involved in oxidative phosphorylation, thermogenesis, biogenesis of iron-sulfur clusters, heme, lipids and amino acids biosynthesis [15-17]; it can modulate programmed cell death [18, 19] and control inflammation [20] as well.”. In the revised clean version lane 39-42.
Reviewer 1 Comment 5
Several specific references are missing. For example, in lane 64 authors describe the involvement of VDAC1 in cancer and neurodegeneration as well as the use of VDAC as a therapeutic target. Authors should briefly describe it in detail and cite the appropriate reference. Also, what is the reference for the sentence in lane 117-119?
Response: Thank you for the comment. The manuscript had been revised.
Information on VDAC1 as a therapeutic target:
“VBIT-3 and VBIT-4, as well as VBIT-12, were reported interacting with VDAC1 by disruption of its oligomerization that results in interfering intracellular Ca2+ concentration, decreasing ROS levels, protecting mitochondrial malfunction related with apoptosis and inflammation [21, 22]. Those were found to alleviate type 2 diabetes [23], lupus [24], atrial myocardium fibrosis [25], ulcerative colitis [26], and amyotrophic lateral sclerosis [22]. Additionally, silencing VDAC1 in cancer cells can suppress tumor cell proliferation in vivo and in vitro [27, 28]; and silencing VDAC1 enhanced mitochondrial function and synaptic activity that provides potential therapeutic approach of neuron related diseases (Alzheimer’s etc.) [29, 30]. The increasing evidences indicate targeting VDAC1 with small molecules might be promising approach worth further in depth investigation that could provide novel strategies against mitochondrial disorder associated diseases.” In the revised clean version lane 173-184.
The reference for the sentence in lane 117-119 (initial submission):
The reference had been updated. “The outer mitochondrial membrane pore gradually enlarges after MOMP, further causing extrusion and rupture of the inner mitochondrial membrane (IMM) which leads to the releasing of mitochondrial DNA (mtDNA) into the cytoplasm [31] (Fig. 2).”In the revised clean version lane 141-144.
Reviewer 1 Comment 6
Lane 94: I’m not sure that MOM can be considered a part of the MPTP; it is true that VDAC1 is located in the MOM, but the other component, ANT, is in the inner membrane. Is this mean that also the IMM belongs to the MPTP? Please revise the sentence.
Response: Thank you for the comment. The authors agree with this concern from the reviewer. The summary in our initial submission from the literature “Even though the structure of the mPTP is not well characterized, it is thought that the mPTP is composed of the outer mitochondrial membrane, a voltage-dependent anion channel (VDAC), and adenine nucleotide translocase (ANT). /doi: 10.3390/ijms22137030./”; and “Adenine Nucleotide Translocator (ANT) and possibly the Voltage Dependent Anion Channel (VDAC) to form the MPT pore. / doi: 10.1016/j.bbagen.2014.11.009./”. Additionally, one more reference had been added to support this statement /Jia K, Du H. Mitochondrial Permeability Transition: A Pore Intertwines Brain Aging and Alzheimer's Disease. Cells. 2021 Mar 15;10(3):649. doi: 10.3390/cells10030649. PMID: 33804048; PMCID: PMC8001058./.
To make the statement clearer, the sentence had been revised to “The voltage-dependent anion channel (VDAC) is located in the outer mitochondrial membrane (OMM); adenine nucleotide translocase (ANT) is located in the inner mitochondrial membrane (IMM). VDAC and ANT are considered to be the structural components of the MPTP [32-34].” In the revised clean version lane 123-126.
Reviewer 1 Comment 7
Lane 341: the passage of metabolites across VDAC1 is bi-directional. Authors should clearly state that while ADP and NAD+ are normally imported into mitochondria, ATP and NADH are mainly exported. Thus, using the word “into” is not correct.
Response: Thank you for the helpful comment. The original statement “VDAC1 affects metabolism by mediating the transport of metabolites such as pyruvate, propionate, succinate, adenosine triphosphate (ATP) and adenosine diphosphate (ADP), and nicotinamide adenine dinucleotide hydrogen (NAD+/NADH) into the mitochondria.” had been revised to “VDAC1 affects metabolism by mediating the transport of metabolites such as pyruvate, propionate, succinate, adenosine triphosphate (ATP) and adenosine diphosphate (ADP), and nicotinamide adenine dinucleotide hydrogen (NAD+/NADH) across the mitochondria, while ADP and NAD+ are normally imported into mitochondria, ATP and NADP are mainly exported into the cytoplasm [35].”. In the revised clean version lane 376-380.
Reviewer 1 Comment 8
Lane 346: authors haven’t take in consideration a relative work recently published by De Pinto group in which it has been demonstrated that VDAC1 knock-out not only decrease metabolites exchanges across the mitochondria, but also it promote a complete rewiring of the whole cell metabolism (doi: 10.1007/s00018-019-03342-8). Authors should add these information and cite the appropriate reference.
Response: Thank you for the comment. The manuscript had been revised accordingly.
“Since VDAC1 is a key protein on the outer mitochondrial membrane contributing to metabolite and ion transportation; it’s better to know what might happen if it has been knocked out. Microarray analysis of VDAC1-null strain indicated the expression of mitochondrial genes was completely reprogrammed accompanied with significantly decreasing of mtDNA. To survival, the mitochondrial metabolism was completely re-arranged as TCA cycle turned on the backup pattern to overcome this dysfunction [36]. Notably, VDAC1 inhibitor, such as VBIT-4, did not detect toxicities in vitro and in vivo [21] that suggests it could be promoted to clinical trials for further investigation.” In the revised clean version lane 387-395.
Reviewer 1 Comment 9
I strongly recommend an overall revision of the language and the grammar. Following several examples:
Response: Thank you for the suggestion. We have seriously revised the language and the grammar throughout the manuscript. The co-author Jay Overholser is an American native speaker and with over 30 years of academic experience. We hope the revised manuscript is more readable, without language or grammar issues. We hope we have met your acceptance requirements for publication.
Reviewer 1 Comment 10
Lane 34: “in moderation” should be replaced with “when moderate”.
Response: Thank you for the comment. We corrected it accordingly, “when in moderate”. In the revised clean version lane 34.
Reviewer 1 Comment 11
Lane 35: “that resulting” should be “that results” or “resulting”.
Response: Thank you for the comment. We corrected it accordingly. In the revised clean version lane 35.
Reviewer 1 Comment 12
Lane 66: the semicolon should be removed.
Response: We appreciate the reviewer’s carefully reading. We corrected it accordingly. In the revised clean version lane 95.
Reviewer 1 Comment 13
Lane 85: “It also indicated” should be “It has been also indicated”.
Response: Thank you for the comment. We corrected it accordingly. In the revised clean version lane 114.
Reviewer 1 Comment 14
Lane 117: MOMP is repeated twice; the second should be replaced with “that”.
Response: Thank you for the comment. We corrected it accordingly. In the revised clean version lane 141.
Reviewer 1 Comment 15
Lane 133: replace “and” with a comma.
Response: Thank you for the comment. We corrected it accordingly. In the revised clean version lane 157.
Reviewer 1 Comment 16
Lane 274: the sentence “…in phagocytosis, elevated cytosolic calcium…” should be replaced with “…in phagocytosis; indeed, elevated cytosolic calcium…”
Response: Thank you for the comment. We corrected it accordingly. In the revised clean version lane 310.
Reviewer 1 Comment 17
Lane 290: remove “et al”
Response: Thank you for the comment. We corrected it accordingly. In the revised clean version lane 327.
Reviewer 1 Comment 18
In many cases, the comma before “and” should be removed.
Response: Thank you for the comment. We corrected it accordingly throughout the manuscript. Please kindly check the tracking version for details.
Reference
- Schein SJ, Colombini M, Finkelstein A: Reconstitution in planar lipid bilayers of a voltage-dependent anion-selective channel obtained from paramecium mitochondria. The Journal of membrane biology 1976, 30(2):99-120.
- Di Rosa MC, Guarino F, Conti Nibali S, Magrì A, De Pinto V: Voltage-Dependent Anion Selective Channel Isoforms in Yeast: Expression, Structure, and Functions. Frontiers in physiology 2021, 12:675708.
- Guardiani C, Magrì A, Karachitos A, Di Rosa MC, Reina S, Bodrenko I, Messina A, Kmita H, Ceccarelli M, De Pinto V: yVDAC2, the second mitochondrial porin isoform of Saccharomyces cerevisiae. Biochimica et biophysica acta Bioenergetics 2018, 1859(4):270-279.
- De Stefani D, Bononi A, Romagnoli A, Messina A, De Pinto V, Pinton P, Rizzuto R: VDAC1 selectively transfers apoptotic Ca2+ signals to mitochondria. Cell death and differentiation 2012, 19(2):267-273.
- Messina A, Reina S, Guarino F, De Pinto V: VDAC isoforms in mammals. Biochimica et biophysica acta 2012, 1818(6):1466-1476.
- Zinghirino F, Pappalardo XG, Messina A, Guarino F, De Pinto V: Is the secret of VDAC Isoforms in their gene regulation? Characterization of human VDAC genes expression profile, promoter activity, and transcriptional regulators. International journal of molecular sciences 2020, 21(19).
- Geula S, Ben-Hail D, Shoshan-Barmatz V: Structure-based analysis of VDAC1: N-terminus location, translocation, channel gating and association with anti-apoptotic proteins. The Biochemical journal 2012, 444(3):475-485.
- Hiller S, Abramson J, Mannella C, Wagner G, Zeth K: The 3D structures of VDAC represent a native conformation. Trends in biochemical sciences 2010, 35(9):514-521.
- Bayrhuber M, Meins T, Habeck M, Becker S, Giller K, Villinger S, Vonrhein C, Griesinger C, Zweckstetter M, Zeth K: Structure of the human voltage-dependent anion channel. Proceedings of the National Academy of Sciences of the United States of America 2008, 105(40):15370-15375.
- Hiller S, Garces RG, Malia TJ, Orekhov VY, Colombini M, Wagner G: Solution structure of the integral human membrane protein VDAC-1 in detergent micelles. Science (New York, NY) 2008, 321(5893):1206-1210.
- Ujwal R, Cascio D, Colletier JP, Faham S, Zhang J, Toro L, Ping P, Abramson J: The crystal structure of mouse VDAC1 at 2.3 A resolution reveals mechanistic insights into metabolite gating. Proceedings of the National Academy of Sciences of the United States of America 2008, 105(46):17742-17747.
- Cheng EH, Sheiko TV, Fisher JK, Craigen WJ, Korsmeyer SJ: VDAC2 inhibits BAK activation and mitochondrial apoptosis. Science (New York, NY) 2003, 301(5632):513-517.
- Reina S, Nibali SC, Tomasello MF, Magrì A, Messina A, De Pinto V: Voltage Dependent Anion Channel 3 (VDAC3) protects mitochondria from oxidative stress. Redox biology 2022, 51:102264.
- Zinghirino F, Pappalardo XG, Messina A, Nicosia G, De Pinto V, Guarino F: VDAC Genes Expression and Regulation in Mammals. Frontiers in physiology 2021, 12:708695.
- Hengartner MO: The biochemistry of apoptosis. Nature 2000, 407(6805):770-776.
- Nunnari J, Suomalainen A: Mitochondria: in sickness and in health. Cell 2012, 148(6):1145-1159.
- Pfanner N, Warscheid B, Wiedemann N: Mitochondrial proteins: from biogenesis to functional networks. Nature reviews Molecular cell biology 2019, 20(5):267-284.
- Bock FJ, Tait SWG: Mitochondria as multifaceted regulators of cell death. 2020, 21(2):85-100.
- Tait SW, Green DR: Mitochondrial regulation of cell death. Cold Spring Harbor perspectives in biology 2013, 5(9).
- Marchi S, Guilbaud E, Tait SWG, Yamazaki T, Galluzzi L: Mitochondrial control of inflammation. Nature reviews Immunology 2022:1-15.
- Ben-Hail D, Begas-Shvartz R, Shalev M, Shteinfer-Kuzmine A, Gruzman A, Reina S, De Pinto V, Shoshan-Barmatz V: Novel Compounds Targeting the Mitochondrial Protein VDAC1 Inhibit Apoptosis and Protect against Mitochondrial Dysfunction. The Journal of biological chemistry 2016, 291(48):24986-25003.
- Shteinfer-Kuzmine A, Argueti-Ostrovsky S, Leyton-Jaimes MF, Anand U, Abu-Hamad S, Zalk R, Shoshan-Barmatz V, Israelson A: Targeting the Mitochondrial Protein VDAC1 as a Potential Therapeutic Strategy in ALS. International journal of molecular sciences 2022, 23(17).
- Zhang E, Mohammed Al-Amily I, Mohammed S, Luan C, Asplund O, Ahmed M, Ye Y, Ben-Hail D, Soni A, Vishnu N et al: Preserving Insulin Secretion in Diabetes by Inhibiting VDAC1 Overexpression and Surface Translocation in β Cells. Cell metabolism 2019, 29(1):64-77.e66.
- Kim J, Gupta R, Blanco LP, Yang S, Shteinfer-Kuzmine A, Wang K, Zhu J, Yoon HE, Wang X, Kerkhofs M et al: VDAC oligomers form mitochondrial pores to release mtDNA fragments and promote lupus-like disease. Science (New York, NY) 2019, 366(6472):1531-1536.
- Klapper-Goldstein H, Verma A, Elyagon S, Gillis R, Murninkas M, Pittala S, Paul A, Shoshan-Barmatz V, Etzion Y: VDAC1 in the diseased myocardium and the effect of VDAC1-interacting compound on atrial fibrosis induced by hyperaldosteronism. Scientific reports 2020, 10(1):22101.
- Verma A, Pittala S, Alhozeel B, Shteinfer-Kuzmine A, Ohana E, Gupta R, Chung JH, Shoshan-Barmatz V: The role of the mitochondrial protein VDAC1 in inflammatory bowel disease: a potential therapeutic target. Molecular therapy : the journal of the American Society of Gene Therapy 2022, 30(2):726-744.
- Shoshan-Barmatz V, Ben-Hail D, Admoni L, Krelin Y, Tripathi SS: The mitochondrial voltage-dependent anion channel 1 in tumor cells. Biochimica et biophysica acta 2015, 1848(10 Pt B):2547-2575.
- Arif T, Vasilkovsky L, Refaely Y, Konson A, Shoshan-Barmatz V: Silencing VDAC1 Expression by siRNA Inhibits Cancer Cell Proliferation and Tumor Growth In Vivo. Molecular therapy Nucleic acids 2017, 8:493.
- Manczak M, Reddy PH: RNA silencing of genes involved in Alzheimer's disease enhances mitochondrial function and synaptic activity. Biochimica et biophysica acta 2013, 1832(12):2368-2378.
- Smilansky A, Dangoor L, Nakdimon I, Ben-Hail D, Mizrachi D, Shoshan-Barmatz V: The Voltage-dependent Anion Channel 1 Mediates Amyloid β Toxicity and Represents a Potential Target for Alzheimer Disease Therapy. The Journal of biological chemistry 2015, 290(52):30670-30683.
- Riley JS, Quarato G, Cloix C, Lopez J, O'Prey J, Pearson M, Chapman J, Sesaki H, Carlin LM, Passos JF et al: Mitochondrial inner membrane permeabilisation enables mtDNA release during apoptosis. The EMBO journal 2018, 37(17).
- Moya GE, Rivera PD, Dittenhafer-Reed KE: Evidence for the Role of Mitochondrial DNA Release in the Inflammatory Response in Neurological Disorders. International journal of molecular sciences 2021, 22(13).
- Gutiérrez-Aguilar M, Baines CP: Structural mechanisms of cyclophilin D-dependent control of the mitochondrial permeability transition pore. Biochimica et biophysica acta 2015, 1850(10):2041-2047.
- Jia K, Du H: Mitochondrial Permeability Transition: A Pore Intertwines Brain Aging and Alzheimer's Disease. Cells 2021, 10(3).
- Shoshan-Barmatz V, Shteinfer-Kuzmine A, Verma A: VDAC1 at the Intersection of Cell Metabolism, Apoptosis, and Diseases. Biomolecules 2020, 10(11).
- Magrì A, Di Rosa MC, Orlandi I, Guarino F, Reina S, Guarnaccia M, Morello G, Spampinato A, Cavallaro S, Messina A et al: Deletion of Voltage-Dependent Anion Channel 1 knocks mitochondria down triggering metabolic rewiring in yeast. Cellular and molecular life sciences : CMLS 2020, 77(16):3195-3213.

Reviewer 2 Report
In this manuscript, the authors addressed the mitochondrial protein VDAC1 a potential therapeutic target of inflammation diseases and possible clinical opportunities. In general, this review summarizes the involvement of VDAC in inflammation and covering the topic related published work. It presents many interesting findings and discoveries, presents new concepts and ideas, and encourage studies to explore.
However, the following comments should be addressed before the manuscript can be accepted for publication.
Comments
1. This MS presents and highly appreciates the functions of VDAC1 in cell functions and proposed and suggest that it is a reasonable target for developing the next generation therapeutic drugs. However, the newly developed VDAC-interacting molecules VBIT-4 and VBIT-12 (J. Biol. Chem. 291(48):24986-25003), inhibiting VDAC1 oligomerization, apoptosis, inflammation and were found to alleviates Type 2 diabetes (Cell Metabolism 29, 64-77, 2019), lupus (Science 366, 1531-1536, 2019), cardiac (Scientific Reports 10: https://doi.org/10.1038), and Colitis (Molecular Therapy, 2;30:726-744, 2022) and ALS (Int. J. Mol. Sci. 2022, 23, 9946.).
I suggest presenting these highly promising compounds.
2. There is no strong and clear evidence showing that VDAC1 is ubiquitinated. The model in Fig. 4 suggesting ubiquitination of VDAC1 by Parkin. This suggestion is most probably based mainly on a single publication from 2010. It should be noted that in this and other related studies VDAC ubiquitination was observed only when PARKIN was expressed but not by endogenous parkin. Thus, this should be clearly indicated, unless this reviewer missed other studies, then they should be indicated.
3. Fig. 4B, Mitophagy interacts with NLRP3, MAVS, ROS, and mtDNA affects immune response.
Not clear how mitophagy can interacts with these proteins/factors? may be modulated, controlled?
4. Why the authors define the following enzymes pyruvate dehydrogenase, isocitrate dehydrogenase, and α-ketoglutarate dehydrogenase, as pro-inflammatory, extend this.
5. Similarly, how the metabolism of phosphoenolpyruvate, lactic acid, succinic acid, citric acid, etc., plays an important role in the occurrence and development of inflammation? Extend
Minor comments
1. Abstract - VDAC is not producing energy but regulation energy production, correct
2. Ref 19 and Ref 70 are identical
3. The question presented: ”Which regions of the protein involved in the VDAC1 oligomerization” has been addressed by Geula et al, JBC, 2012, and have identified the b-strands involve in the oligomerization.
4. It is indicated that “none of VDAC1 clinical trial has been conducted yet to the best of our knowledge”. There is 3 companies established around VDAC1, one of which, ViDAC Pharma has conducted clinical trials on VDAC1 interacting molecule for cancer treatment .
5. Inner mitochondrial membrane (IMM) and not mitochondria inner membrane (MIM) as used here, I suggest changing this.
Author Response
Reviewer 2
In this manuscript, the authors addressed the mitochondrial protein VDAC1 a potential therapeutic target of inflammation diseases and possible clinical opportunities. In general, this review summarizes the involvement of VDAC in inflammation and covering the topic related published work. It presents many interesting findings and discoveries, presents new concepts and ideas, and encourage studies to explore.
Response: Thank you for your review of our manuscript. The authors appreciate the constructive comments that have greatly helped us improve our manuscript. We have seriously revised the manuscript, please kindly check the tracking version and the clean version marked in red. The following are the point by point response.
However, the following comments should be addressed before the manuscript can be accepted for publication.
Reviewer 2 Major Comment 1
This MS presents and highly appreciates the functions of VDAC1 in cell functions and proposed and suggest that it is a reasonable target for developing the next generation therapeutic drugs. However, the newly developed VDAC-interacting molecules VBIT-4 and VBIT-12 (J. Biol. Chem. 291(48):24986-25003), inhibiting VDAC1 oligomerization, apoptosis, inflammation and were found to alleviates Type 2 diabetes (Cell Metabolism 29, 64-77, 2019), lupus (Science 366, 1531-1536, 2019), cardiac (Scientific Reports 10: https://doi.org/10.1038), and Colitis (Molecular Therapy, 2;30:726-744, 2022) and ALS (Int. J. Mol. Sci. 2022, 23, 9946.). I suggest presenting these highly promising compounds.
Response: Thank you for the valuable comments. The manuscript had been revised.
“VBIT-3 and VBIT-4, as well as VBIT-12, were reported interacting with VDAC1 by disruption of its oligomerization that results in interfering intracellular Ca2+ concentration, decreasing ROS levels, protecting mitochondrial malfunction related with apoptosis and inflammation [1, 2]. Those were found to alleviate type 2 diabetes [3], lupus [4], atrial myocardium fibrosis [5], ulcerative colitis [6], and amyotrophic lateral sclerosis [2]. Additionally, silencing VDAC1 in cancer cells can suppress tumor cell proliferation in vivo and in vitro [7, 8]; and silencing VDAC1 enhanced mitochondrial function and synaptic activity that provides potential therapeutic approach of neuron related diseases (Alzheimer’s etc.) [9, 10]. The increasing evidences indicate targeting VDAC1 with small molecules might be promising approach that worth further in depth investigation that could provide novel strategies against mitochondrial disorder associated diseases.” In the revised clean version lane 173-184.
Reviewer 2 Major Comment 2
There is no strong and clear evidence showing that VDAC1 is ubiquitinated. The model in Fig. 4 suggesting ubiquitination of VDAC1 by Parkin. This suggestion is most probably based mainly on a single publication from 2010. It should be noted that in this and other related studies VDAC ubiquitination was observed only when PARKIN was expressed but not by endogenous parkin. Thus, this should be clearly indicated, unless this reviewer missed other studies, then they should be indicated.
Response: Thank you for the comment. The original section 6 had been reconstructed and seriously revised. The related studies of VDAC ubiquitination have been updated. In the revised clean version lane 546-567.
“The PTEN-induced putative kinase 1 (PINK1) and the RING family ubiquitin ligase Parkin were found to be involved in mitophagy [11-14]. This indicates that induced mitophagy can be accomplished in cells overexpressing Parkin or overexpressing PINK1. PINK1/Parkin acts as key regulator of mitophagy and vital in controlling infection and inflammation response [15]. The interaction of two Parkin domains, RING1 and Ubiquitin-like (UBL), affects its activity. UBL binding with RING1 results in the inactive state of Parkin; PINK1 phosphorylates UBL-Ser65 leading to the activation of Parkin to promote substrate ubiquitination, VDAC1 is included [16].
Studies have demonstrated that Parkin interacts with VDACs, and VDAC1 is the target of Parkin-mediated Lys27 polyubiquitination and mitochondrial phagocytosis [17-20]. VDACs are effective in helping Parkin indentify defective mitochondria and assist in subsequent mitochondrial phagocytosis, and VDAC1 is necessary for PINK1/Parkin targeting damaged mitochondria [17]. Partial silencing of the VDAC1 resulted in significantly reduced Parkin translocation from cytoplasm to damaged mitochondria while also significantly preventing mitochondrial clearance, notably retransfection of flag-tagged VDAC1 significantly restored Parkin mitochondrial translocation and clearance [17]. Parkin ubiquitinates VDAC1 and ultimately selectively degrades damaged mitochondria by promoting mitophagy [20]. Notably, the ubiquitylation of VDAC1 was observed at enhanced expression of Parkin instead of endogenous Parkin [17]. It has shown that Parkin's targeting of defective mitochondria is impaired in the absence of both VDACs, but can be rescued by expressing VDAC1 or VDAC3 in these cells [20]. These pieces of evidence confirm that VDAC1 is important for PINK1/Parkin-involved mitophagy (Fig. 4A).”.
Reviewer 2 Major Comment 3
Fig. 4B, Mitophagy interacts with NLRP3, MAVS, ROS, and mtDNA affects immune response. Not clear how mitophagy can interacts with these proteins/factors? may be modulated, controlled?
Response: Thank you for the comment. The incorrect description word has been rephrased accordingly.
“Mitophagy modulates NLRP3, MAVS and mtDNA release affects immune response.”In the revised clean version lane 662.
Description of “mitophagy and NLRP3”: In the revised clean version lane 594-604.
Description of “mitophagy and MAVS”: In the revised clean version lane 577-585.
Description of “mitophagy and mtDNA release”: In the revised clean version lane 608-657, section 6.2 mitophagy modulates mtDNA level in cytoplasm.
Reviewer 2 Major Comment 4
Why the authors define the following enzymes pyruvate dehydrogenase, isocitrate dehydrogenase, and α-ketoglutarate dehydrogenase, as pro-inflammatory, extend this.
Response: Thank you for the comment. The description is located on V1 version lane 425-428 “VDAC1 can affect various respiratory enzymes involved in mitochondrial respiration as well as pro-inflammatory substances, including pyruvate dehydrogenase, isocitrate de-hydrogenase, and α-ketoglutarate dehydrogenase, by affecting the transport of mitochondrial metabolites.” Sorry for the misleading description, that’s not what we would like to express. We rephrased the description to “VDAC1 can affect mitochondrial respiration resulting from its important role in controlling the transportation of substances and metabolites. The intermediates in the Kreb cycle have close relation with inflammation process [21].” In the revised clean version lane 485-487.
Reviewer 2 Major Comment 5
Similarly, how the metabolism of phosphoenolpyruvate, lactic acid, succinic acid, citric acid, etc., plays an important role in the occurrence and development of inflammation? Extend
Response: Thank you for the constructive comment. To make it clearer, we added more references and extended the relation description accordingly.
“The metabolism of PEP, lactic acid, succinic acid, citric acid, etc., plays an important role in the occurrence and development of inflammation.” Each metabolite has been updated and extended in the previous description.
Phosphoenolpyruvate (PEP)
“Phosphoenolpyruvate (PEP) is produced by enolase-1 during glycolysis and accumulates in T cells, the accumulation of PEP has a similar pro-inflammatory effect on macrophages, promoting M1 poles, which increases the expression of pro-inflammatory cytokines [22, 23]. PEP is associated with inflammation via impacting Ca2+ [22]. PEP can inhibit the ER calcium channel to suppress Ca2+ flux to ER [22] resulting in the increased cytoplasm Ca2+ promotes the activation of nuclear factor of activated T-cells [22, 24].”. In the revised clean version lane 456-461.
Lactic acid
“Lactic acid, the final product of glycolysis, can display signaling properties during inflammation [25]. During this process, lactate suppresses immune responses by impairing the shift of metabolic reorganization to a pro-inflammatory phenotype and blocking pro-inflammatory signaling pathways in monocytes, macrophages and DCs [26, 27]. The accumulation of lactate in DCs drives the switch to an anti-inflammatory phenotype by increasing IL-10 [28]. However, lactate-rich environments have been reported to enhance TH17 responses in macrophages [29]. Lactate can promote Th17 responses and activate NF-κB pro-inflammatory signaling of macrophages [29, 30]. Lactate is able to enter cells, stimulate the NF-κB/IL-8 pathway and induce ROS production [31]. Lactate plays a key role in regulation macrophage polarization, modification of histones and inflammatory response [32, 33]; it also enhances IFN-γ expression and the differentiation of T helper 1 cell [34]. The various role of lactate in inflammation processes have been documented early this year [35].”. In the revised clean version lane 462-474.
Succinic acid
“Succinate accumulation leads to macrophage M1 polarization through direct inhibition of proline hydrolase, prompting HIF-1α and IL-1β secretion [36, 37]; it acts as an inflammation stimulator showing autocrine dependent manner [36, 38]. Lipopolysaccharide (LPS) induced succinate promotes IL-1β expression via HIF-1α signaling [37, 39]. Extracellular succinate induces a pro-inflammatory response in diverse immune cells, increasing the migration and secretion of the pro-inflammatory cytokines TNF-α and IL-1β in dendritic cells and macrophages [37].”. In the revised clean version lane 408-414.
Citric acid
“Citric acid accumulates in LPS-stimulated macrophages [37, 40, 41]; autocrine type I IFN-driven IL-10 to suppress the activity of isocitrate dehydrogenase (IDH) of LPS-treated macrophages to promote this process [40]. Citrate is generated during the tricarboxylic acid reaction and, once in the cytoplasm, is metabolized by ATP-citrate lyase (ACLY) to acetyl-CoA and oxaloacetate, precursors for lipid synthesis, ROS and NO [42]. Citrate effects ICAM-1 and cytokine (eg. IL-6) contributing to the regulation of endothelial inflammation [43]; it acts as an anti-inflammation factor [44, 45]. Studies have suggested that reduced cytoplasmic citrate levels due to depletion of circulating immune complexes (CICs) reduce ROS, NO and prostaglandin production. These changes may impair pro-inflammatory differentiation of cells, underscoring the role of certain metabolites in the inflammatory response in the importance [22, 42].”. In the revised clean version lane 415-425.
Minor comments
Reviewer 2 Minor Comment 1
Abstract - VDAC is not producing energy but regulation energy production, correct
Response: Thank you for the comment. That’s what we want to express. The word “regulating” is in the front of the sentence. To make it clearer, the description has been modified, “VDAC1 is involved in regulating energy production, mitochondrial oxidase stress, Ca2+ trans-portation, substance metabolism, apoptosis, mitochondrial autophagy (mitophagy), and many other functions.” In the revised clean version lane 15-17.
Reviewer 2 Minor Comment 2
Ref 19 and Ref 70 are identical
Response: Thank you for your carefully reading. The citations had been updated.
Reviewer 2 Minor Comment 3
The question presented: ”Which regions of the protein involved in the VDAC1 oligomerization” has been addressed by Geula et al, JBC, 2012, and have identified the b-strands involve in the oligomerization.
Response: Thank you for the comment. The statement had been rephrased. “Although the high-resolution structure of recombinant VDAC1 has been determined, and VDAC1 β-strands had been identified include dimerization sites, Ile-27, Leu-29, Thr-51 and Leu-227 are involved in VDAC1 oligomerization [46]. Deep insights mechanism regulation still needs further investigation.” In the revised clean version lane 698-702.
Reviewer 2 Minor Comment 4
It is indicated that “none of VDAC1 clinical trial has been conducted yet to the best of our knowledge”. There is 3 companies established around VDAC1, one of which, ViDAC Pharma has conducted clinical trials on VDAC1 interacting molecule for cancer treatment.
Response: Thank you for the helpful information. The manuscript had been updated. “VDA-1102 was designed to modulate VDAC/HK2 which effects glycolysis and mitochon-drial function in cancer and activated immune cells. VDA-1102 related clinical trials had been conducted against solid tumors by VidacPharma, there is no serious adverse events had been noticed from a Phase II B (NCT 03538951) study (http://www.vidacpharma.com/clinical-trials).” In the revised clean version lane 706-711.
Reviewer 2 Minor Comment 5
Inner mitochondrial membrane (IMM) and not mitochondria inner membrane (MIM) as used here, I suggest changing this.
Response: Thank you for the comment. We are now only use “Inner mitochondrial membrane (IMM)” or “Outer mitochondrial membrane (OMM)” throughout the manuscript.
Reference
- Ben-Hail D, Begas-Shvartz R, Shalev M, Shteinfer-Kuzmine A, Gruzman A, Reina S, De Pinto V, Shoshan-Barmatz V: Novel Compounds Targeting the Mitochondrial Protein VDAC1 Inhibit Apoptosis and Protect against Mitochondrial Dysfunction. The Journal of biological chemistry 2016, 291(48):24986-25003.
- Shteinfer-Kuzmine A, Argueti-Ostrovsky S, Leyton-Jaimes MF, Anand U, Abu-Hamad S, Zalk R, Shoshan-Barmatz V, Israelson A: Targeting the Mitochondrial Protein VDAC1 as a Potential Therapeutic Strategy in ALS. International journal of molecular sciences 2022, 23(17).
- Zhang E, Mohammed Al-Amily I, Mohammed S, Luan C, Asplund O, Ahmed M, Ye Y, Ben-Hail D, Soni A, Vishnu N et al: Preserving Insulin Secretion in Diabetes by Inhibiting VDAC1 Overexpression and Surface Translocation in β Cells. Cell metabolism 2019, 29(1):64-77.e66.
- Kim J, Gupta R, Blanco LP, Yang S, Shteinfer-Kuzmine A, Wang K, Zhu J, Yoon HE, Wang X, Kerkhofs M et al: VDAC oligomers form mitochondrial pores to release mtDNA fragments and promote lupus-like disease. Science (New York, NY) 2019, 366(6472):1531-1536.
- Klapper-Goldstein H, Verma A, Elyagon S, Gillis R, Murninkas M, Pittala S, Paul A, Shoshan-Barmatz V, Etzion Y: VDAC1 in the diseased myocardium and the effect of VDAC1-interacting compound on atrial fibrosis induced by hyperaldosteronism. Scientific reports 2020, 10(1):22101.
- Verma A, Pittala S, Alhozeel B, Shteinfer-Kuzmine A, Ohana E, Gupta R, Chung JH, Shoshan-Barmatz V: The role of the mitochondrial protein VDAC1 in inflammatory bowel disease: a potential therapeutic target. Molecular therapy : the journal of the American Society of Gene Therapy 2022, 30(2):726-744.
- Shoshan-Barmatz V, Ben-Hail D, Admoni L, Krelin Y, Tripathi SS: The mitochondrial voltage-dependent anion channel 1 in tumor cells. Biochimica et biophysica acta 2015, 1848(10 Pt B):2547-2575.
- Arif T, Vasilkovsky L, Refaely Y, Konson A, Shoshan-Barmatz V: Silencing VDAC1 Expression by siRNA Inhibits Cancer Cell Proliferation and Tumor Growth In Vivo. Molecular therapy Nucleic acids 2017, 8:493.
- Manczak M, Reddy PH: RNA silencing of genes involved in Alzheimer's disease enhances mitochondrial function and synaptic activity. Biochimica et biophysica acta 2013, 1832(12):2368-2378.
- Smilansky A, Dangoor L, Nakdimon I, Ben-Hail D, Mizrachi D, Shoshan-Barmatz V: The Voltage-dependent Anion Channel 1 Mediates Amyloid β Toxicity and Represents a Potential Target for Alzheimer Disease Therapy. The Journal of biological chemistry 2015, 290(52):30670-30683.
- Matsuda N, Sato S, Shiba K, Okatsu K, Saisho K, Gautier CA, Sou YS, Saiki S, Kawajiri S, Sato F et al: PINK1 stabilized by mitochondrial depolarization recruits Parkin to damaged mitochondria and activates latent Parkin for mitophagy. The Journal of cell biology 2010, 189(2):211-221.
- Vives-Bauza C, Zhou C, Huang Y, Cui M, de Vries RL, Kim J, May J, Tocilescu MA, Liu W, Ko HS et al: PINK1-dependent recruitment of Parkin to mitochondria in mitophagy. Proceedings of the National Academy of Sciences of the United States of America 2010, 107(1):378-383.
- Shiba-Fukushima K, Imai Y, Yoshida S, Ishihama Y, Kanao T, Sato S, Hattori N: PINK1-mediated phosphorylation of the Parkin ubiquitin-like domain primes mitochondrial translocation of Parkin and regulates mitophagy. Scientific reports 2012, 2:1002.
- Narendra D, Tanaka A, Suen DF, Youle RJ: Parkin is recruited selectively to impaired mitochondria and promotes their autophagy. The Journal of cell biology 2008, 183(5):795-803.
- Cho DH, Kim JK, Jo EK: Mitophagy and Innate Immunity in Infection. Molecules and cells 2020, 43(1):10-22.
- Ham SJ, Lee SY, Song S, Chung JR, Choi S, Chung J: Interaction between RING1 (R1) and the Ubiquitin-like (UBL) Domains Is Critical for the Regulation of Parkin Activity. The Journal of biological chemistry 2016, 291(4):1803-1816.
- Geisler S, Holmström KM, Skujat D, Fiesel FC, Rothfuss OC, Kahle PJ, Springer W: PINK1/Parkin-mediated mitophagy is dependent on VDAC1 and p62/SQSTM1. Nature cell biology 2010, 12(2):119-131.
- Ham SJ, Lee D, Yoo H, Jun K, Shin H, Chung J: Decision between mitophagy and apoptosis by Parkin via VDAC1 ubiquitination. Proceedings of the National Academy of Sciences of the United States of America 2020, 117(8):4281-4291.
- Yang X, Zhou Y, Liang H, Meng Y, Liu H, Zhou Y, Huang C, An B, Mao H, Liao Z: VDAC1 promotes cardiomyocyte autophagy in anoxia/reoxygenation injury via the PINK1/Parkin pathway. Cell biology international 2021, 45(7):1448-1458.
- Sun Y, Vashisht AA, Tchieu J, Wohlschlegel JA, Dreier L: Voltage-dependent anion channels (VDACs) recruit Parkin to defective mitochondria to promote mitochondrial autophagy. The Journal of biological chemistry 2012, 287(48):40652-40660.
- Mills EL, Kelly B, O'Neill LAJ: Mitochondria are the powerhouses of immunity. Nature immunology 2017, 18(5):488-498.
- Soto-Heredero G, Gómez de Las Heras MM, Gabandé-Rodríguez E, Oller J, Mittelbrunn M: Glycolysis - a key player in the inflammatory response. The FEBS journal 2020, 287(16):3350-3369.
- Vander Heiden MG, Locasale JW, Swanson KD, Sharfi H, Heffron GJ, Amador-Noguez D, Christofk HR, Wagner G, Rabinowitz JD, Asara JM et al: Evidence for an alternative glycolytic pathway in rapidly proliferating cells. Science (New York, NY) 2010, 329(5998):1492-1499.
- Ho PC, Bihuniak JD, Macintyre AN, Staron M, Liu X, Amezquita R, Tsui YC, Cui G, Micevic G, Perales JC et al: Phosphoenolpyruvate Is a Metabolic Checkpoint of Anti-tumor T Cell Responses. Cell 2015, 162(6):1217-1228.
- Brooks GA: The Science and Translation of Lactate Shuttle Theory. Cell metabolism 2018, 27(4):757-785.
- Pearce EJ, Everts B: Dendritic cell metabolism. Nature reviews Immunology 2015, 15(1):18-29.
- Errea A, Cayet D, Marchetti P, Tang C, Kluza J, Offermanns S, Sirard JC, Rumbo M: Lactate Inhibits the Pro-Inflammatory Response and Metabolic Reprogramming in Murine Macrophages in a GPR81-Independent Manner. PloS one 2016, 11(11):e0163694.
- Nasi A, Fekete T, Krishnamurthy A, Snowden S, Rajnavölgyi E, Catrina AI, Wheelock CE, Vivar N, Rethi B: Dendritic cell reprogramming by endogenously produced lactic acid. Journal of immunology (Baltimore, Md : 1950) 2013, 191(6):3090-3099.
- Haas R, Smith J, Rocher-Ros V, Nadkarni S, Montero-Melendez T, D'Acquisto F, Bland EJ, Bombardieri M, Pitzalis C, Perretti M et al: Lactate Regulates Metabolic and Pro-inflammatory Circuits in Control of T Cell Migration and Effector Functions. PLoS biology 2015, 13(7):e1002202.
- Samuvel DJ, Sundararaj KP, Nareika A, Lopes-Virella MF, Huang Y: Lactate boosts TLR4 signaling and NF-kappaB pathway-mediated gene transcription in macrophages via monocarboxylate transporters and MD-2 up-regulation. Journal of immunology (Baltimore, Md : 1950) 2009, 182(4):2476-2484.
- Végran F, Boidot R, Michiels C, Sonveaux P, Feron O: Lactate influx through the endothelial cell monocarboxylate transporter MCT1 supports an NF-κB/IL-8 pathway that drives tumor angiogenesis. Cancer research 2011, 71(7):2550-2560.
- Ivashkiv LB: The hypoxia-lactate axis tempers inflammation. Nature reviews Immunology 2020, 20(2):85-86.
- Colegio OR, Chu NQ, Szabo AL, Chu T, Rhebergen AM, Jairam V, Cyrus N, Brokowski CE, Eisenbarth SC, Phillips GM et al: Functional polarization of tumour-associated macrophages by tumour-derived lactic acid. Nature 2014, 513(7519):559-563.
- Peng M, Yin N, Chhangawala S, Xu K, Leslie CS, Li MO: Aerobic glycolysis promotes T helper 1 cell differentiation through an epigenetic mechanism. Science (New York, NY) 2016, 354(6311):481-484.
- Manosalva C, Quiroga J, Hidalgo AI, Alarcón P, Anseoleaga N, Hidalgo MA, Burgos RA: Role of Lactate in Inflammatory Processes: Friend or Foe. Frontiers in immunology 2021, 12:808799.
- Viola A, Munari F, Sánchez-Rodríguez R, Scolaro T, Castegna A: The Metabolic Signature of Macrophage Responses. Frontiers in immunology 2019, 10:1462.
- Tannahill GM, Curtis AM, Adamik J, Palsson-McDermott EM, McGettrick AF, Goel G, Frezza C, Bernard NJ, Kelly B, Foley NH et al: Succinate is an inflammatory signal that induces IL-1β through HIF-1α. Nature 2013, 496(7444):238-242.
- Littlewood-Evans A, Sarret S, Apfel V, Loesle P, Dawson J, Zhang J, Muller A, Tigani B, Kneuer R, Patel S et al: GPR91 senses extracellular succinate released from inflammatory macrophages and exacerbates rheumatoid arthritis. The Journal of experimental medicine 2016, 213(9):1655-1662.
- Mills E, O'Neill LA: Succinate: a metabolic signal in inflammation. Trends in cell biology 2014, 24(5):313-320.
- De Souza DP, Achuthan A, Lee MK, Binger KJ, Lee MC, Davidson S, Tull DL, McConville MJ, Cook AD, Murphy AJ et al: Autocrine IFN-I inhibits isocitrate dehydrogenase in the TCA cycle of LPS-stimulated macrophages. The Journal of clinical investigation 2019, 129(10):4239-4244.
- Seim GL, Britt EC, John SV, Yeo FJ, Johnson AR, Eisenstein RS, Pagliarini DJ, Fan J: Two-stage metabolic remodelling in macrophages in response to lipopolysaccharide and interferon-γ stimulation. Nature metabolism 2019, 1(7):731-742.
- Infantino V, Convertini P, Cucci L, Panaro MA, Di Noia MA, Calvello R, Palmieri F, Iacobazzi V: The mitochondrial citrate carrier: a new player in inflammation. The Biochemical journal 2011, 438(3):433-436.
- Bryland A, Wieslander A, Carlsson O, Hellmark T, Godaly G: Citrate treatment reduces endothelial death and inflammation under hyperglycaemic conditions. Diabetes & vascular disease research 2012, 9(1):42-51.
- Williams NC, O'Neill LAJ: A Role for the Krebs Cycle Intermediate Citrate in Metabolic Reprogramming in Innate Immunity and Inflammation. Frontiers in immunology 2018, 9:141.
- Choi EY, Kim HJ, Han JS: Anti-inflammatory effects of calcium citrate in RAW 264.7cells via suppression of NF-κB activation. Environmental toxicology and pharmacology 2015, 39(1):27-34.
- Geula S, Naveed H, Liang J, Shoshan-Barmatz V: Structure-based analysis of VDAC1 protein: defining oligomer contact sites. The Journal of biological chemistry 2012, 287(3):2179-2190.

Round 2
Reviewer 1 Report
Authors have fulfilled my comments. I think the work should be now considered for publication.